# PPDL: LLM-Based Flows as Probabilistic Programs

**Louis Mandel** [1]  **Guillaume Baudart** [2]  **Mandana Vaziri** [1]  **Martin Hirzel** [1]

## Abstract

Building reliable applications that leverage large language models (LLMs) remains a significant challenge. While LLMs offer impressive capabilities across diverse tasks, their outputs often lack accuracy and provide no clear measure of confidence. This uncertainty compounds in flows of multiple calls to LLMs and other tools, making it difficult for developers and end-users to trust the results. This paper introduces a probabilistic language for programming LLM-based flows. It enables developers to quantify and propagate uncertainty throughout the application's flow, and experiment with different inference scaling techniques without adding a single line of code beyond the flow's logic. We present an experimental study to demonstrate this capability, and a case study building a theorem proving agent for the Rocq theorem prover.

## 1. Introduction

While a single call to a large language model (LLM) can perform remarkably well for some tasks, other tasks require flows of multiple calls to LLMs and other tools (Jimenez et al., 2024). But while such LLM-based flows may improve accuracy, unfortunately, they also exacerbate uncertainty. Because each step depends on previous uncertain outputs, uncertainty compounds along the trace of a flow's execution. This makes it difficult for users to assess the correctness of an answer, leaving them wondering how certain it is and what other answers are likely.

Several recent solutions for improving LLM flow accuracy use *inference scaling*, which scales up the number of LLM inference calls or tokens at test time (Chen et al., 2023; Cobbe et al., 2021; Wang et al., 2023; Yao et al., 2023a;b). The idea is to try the same task several times,

[1]IBM, New York, USA [2]Université Paris Cité, Inria, CNRS, IRIF, France. Correspondence to: Louis Mandel <lmandel@us.ibm.com>.

*Proceedings of the $43^{rd}$ International Conference on Machine Learning*, Seoul, South Korea. PMLR 306, 2026. Copyright 2026 by the author(s).

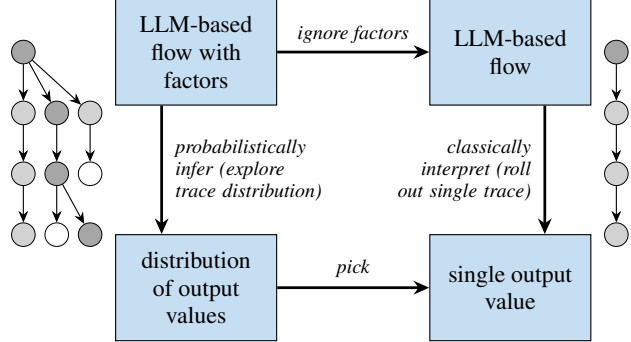

*Figure 1.* Executing LLM-based flows with or without probabilistic programming. Each circle represents an execution state, with darker shades indicating higher normalized probability.

either independently or with feedback between attempts, hoping that one of the outputs is correct. Inference scaling usually leverages *constraints* to refine or select outputs, including soft constraints from models and hard constraints from rule-based tools. Since it explores a distribution of outputs and attempts to shape that distribution via constraints, it could be viewed as a form of probabilistic programming (Gordon et al., 2014). But in practice, it tends to be more ad hoc. While inference scaling could save cost by allocating more resources to better execution traces, that would require a principled way to track multiple partial traces along with their likelihoods. However, current inference scaling implementations are tightly interwoven with the program, increasing complexity and hindering the exploration of alternative paradigms. Furthermore, although in theory inference scaling could help users assess answer quality (since it deals in distributions), in practice it is rarely principled enough to deliver that benefit.

To obtain more accurate results from LLM-based flows while also giving users visibility into their uncertainty, we aim to make inference scaling more principled by incorporating it into a programming language. We introduce *PPDL*, the first probabilistic programming language for LLM-based flows. In general, *Probabilistic programming languages* (Gordon et al., 2014; Carpenter et al., 2017; Goodman et al., 2008; Bingham et al., 2019) decouple the search over a distribution of traces from the core program logic, but they are notoriously hard to use and are not applied to LLM-based flows. *Prompt programming lan-*

guages (Lundberg et al., 2022; Khattab et al., 2024; Zheng et al., 2024; Mell et al., 2025) make it easy to compose flows of LLM and tool calls, but do not help with tracking uncertainty. And while they enable developers to explicitly build ad hoc inference scaling workflows, such close coupling between the search and the core logic is complicated and inflexible.

PPDL's goal is to bring these two paradigms together in a single framework. PPDL is designed for developers and researchers who build multi-step LLM or agentic workflows, including prompt-language users, agent framework developers, and researchers experimenting with inference scaling. PPDL lets users specify the core logic of their program in a high-level prompting language which already provides one of the two core constructs of probabilistic languages, namely *sample* in the form of an LLM call. It extends this language with the second of the two core constructs of probabilistic languages, namely *factor*. A factor updates the probability of the current execution trace based on user-specified soft or hard constraints.

Flow specifications in PPDL require no explicit inference scaling loops, as the runtime takes care of exploring a distribution of possible traces (see Figure 1). Thus, PPDL effectively decouples inference scaling from the core program logic, providing a suite of plug-and-play probabilistic inference engines: simple majority voting, importance sampling, and particle filtering (aka. Sequential Monte Carlo or SMC). Each particle represents a possible trace through the program, which might involve multiple LLM and tool calls. Unlike traditional prompting languages that return a single output, PPDL returns a distribution of outputs. Users can then inspect that distribution to assess uncertainty, or pick a single output, such as the one with the highest probability score.

Besides prompts, sample (LLM calls), and factor, PPDL includes constructs for variables, control structures (loops, conditionals, error handling), functions, file imports, and tool calls. It allows users to write a flow once and then experiment with various inference scaling engines without writing an additional line of code. This paper formalizes a semantics for PPDL that clarifies its behavior as both a prompting language and a probabilistic language, and the resulting distribution of traces and output values. Furthermore, this paper describes an implementation with parallelization both within a single execution trace and across multiple in-flight partial traces using functional data structures. The empirical results show the versatility of PPDL as a framework for experimenting with different inference scaling algorithms across multiple widely used benchmarks, as well as a case study where it is used to implement a theorem proving agent for the Rocq theorem prover (Rocq Prover Team).

This paper makes the following novel contributions:

1. PPDL, the first probabilistic prompt programming language for flows of LLM and tool calls.
2. A semantics that formalizes the interplay between prompt-based sample and probabilistic factor.
3. Results with different probabilistic inference engines across different LLMs and benchmarks.
4. A case study building a theorem proving agent for the Rocq theorem prover.

PPDL is open-source and distributed with PDL: https://github.com/IBM/prompt-declaration-language/tree/icml-26.

**Conflict of Interest Disclosure.** All the authors, except Guillaume Baudart, are employed by IBM, which leads the development of the Granite models, which were among the ones evaluated in this paper.

## 2. Overview

**Prompt programming.** PPDL is based on PDL (Vaziri et al., 2024), a declarative approach to prompt programming that combines human readability with ease of execution. PDL programs represent the composition of calls to LLMs and tools, abstracting away the plumbing necessary for such compositions. PDL is based on the premise that interactions with an LLM are mainly for the purpose of generating data. Users specify the shape of data to be generated in YAML, which makes it easy to see properly formatted prompts. PDL adds enough scripting to allow specifying entire flows. Code blocks allow users to compose LLM calls with arbitrary code written in Python, Jinja2, or Shell. PDL enables implicit accumulation of the context of messages, freeing users from such concerns, and provides type checking and constrained decoding to enforce the shape of LLM outputs.

Consider the example in Figure 2, which illustrates a simple flow: we first ask a model to generate a plan in natural language for how to generate code for a problem, then we chain that with another model call to generate code while taking the plan into account. Lines 1–5 define some variables for use in the rest of the program. Line 6 declares the shape of data to be produced as a `lastOf`, which means a sequence of blocks whose result is that of the last block in the sequence. Lines 7–20 are a list of blocks comprising the body of `lastOf`. Lines 7–9 show a prompt asking for an English plan for a code generation problem. Line 10 shows a declarative `model` call. It provides the id of the model to call from the variable `llm` defined in Line 2 (PDL is based on LiteLLM, so this is a LiteLLM id for a watsonX model). A PDL program implicitly accumulates the context in the background, so the input to this model call is everything that appeared before. In this case, the input to the model is

```
1  defs:
2    llm: watsonx/meta-llama/llama-4-maverick-...
3    problem_statement: |
4      Write a function to find nth centered hexagonal number.
5      assert centered_hexagonal_number(10) == 271
6  lastOf:
7  - >
8    Generate an English plan for how to generate code for
9    the following problem: ${ problem_statement }
10 - model: ${ llm }
11 - >
12   Generate a complete executable Python function
13   definition corresponding to the above plan and problem.
14   Generate only a single function definition.
15 - model: ${ llm }
16   parser:
17     regex: (.|\n)*```python\n(?P(.|\n)*?)```(.|\n)*
18     spec: { code: string }
19   def: solution
20 - ${ solution.code | default("") }
```

*Figure 2.* PDL program generating a Python function.

*Table 1.* (Left) Possible outputs of the PDL code from Figure 2; (Right) Posterior probability and particle count added by Figure 3.

| Output | Probability | Count |
|---|---|---|
| `def centered_hexagonal_number(n):`
`  return 1 + 3 * n * (n + 1)` | 0.00005 | 3 |
| `def centered_hexagonal_number(n):`
`  return 1 + 3 * n * (n - 1)` | 0.99988 | 1 |
| `def centered_hexagonal_number(n):`
`  return 1 + 6 * (n * (n + 1)) // 2` | 0.00007 | 1 |

```
1  defs:
2    llm: watsonx/meta-llama/llama-4-maverick-...
3    problem_statement: |
4      Write a function to find nth centered hexagonal number.
5      assert centered_hexagonal_number(10) == 271
6    utils:
7      import: utils.pdl
8  lastOf:
9  - >
10   Generate an English plan for how to generate code for
11   the following problem: ${ problem_statement }
12 - model: ${ llm }
13   def: plan
14 - defs:
15     constraint: |
16       This plan for the following problem is correct.
17       ${ problem_statement }
18   factor: ${ utils.llm_judge(llm, plan, constraint) }
19 - >
20   Generate a complete executable Python function
21   definition corresponding to the above plan and problem.
22   Generate only a single function definition.
23 - model: ${ llm }
24   parser:
25     regex: (.|\n)*```python\n(?P(.|\n)*?)```(.|\n)*
26     spec: { code: string }
27   def: solution
28 - factor: ${ utils.score_errors_and_warnings(solution) }
29 - ${ solution.code | default("") }
```

*Figure 3.* PPDL program adding constraints with `factor`.

answer with the highest probability, providing a greater likelihood of correctness.

Figure 3 shows the same program as Figure 2, but with added constraints. Lines 6–7 import `utils`, a library with utility functions not shown in this figure. Lines 14–18 score the plan that was generated using an LLM-as-a-judge (Zheng et al., 2023). The function `utils.llm_judge` takes a model name, a response, and a constraint, and checks whether the response meets the constraint. It is an LLM call that returns true or false, and we use the log probabilities output by the LLM to compute a score according to the following formula:

$$\text{score} = \log \left( \frac{\exp(\text{lp}_t)}{\exp(\text{lp}_t) + \exp(\text{lp}_f)} \right)$$

where $\text{lp}_t$ and $\text{lp}_f$ are the log probabilities for *true* and *false*, respectively. Line 28 shows a rule-based constraint, where we compute a score based on the errors and warnings from the flake8 linter.

Each execution of the program now generates an output and a score that reflect the quality of the corresponding execution with respect to the user constraints introduced by `factor`. The PPDL runtime then performs *probabilistic inference*, which leverages these scores to estimate the *posterior distribution* of possible outputs. Probabilistic inference is, in general, intractable, and the distributions computed by LLMs operate over enormous spaces. PPDL thus

the prompt on Lines 7–9. Next, we formulate a prompt to generate code (Lines 11–14). This is followed by another `model` call on Lines 15–19 for code generation. Its input includes all previous prompts and the response of the first model call. The output is parsed using the regular expression on Line 17, which defines the `code` capture group. On Line 18, a type specification enforces the shape of the output to be a JSON object with a `code` field. Line 19 defines the variable `solution` to hold the response of the LLM. Finally, Line 20 contains a Jinja expression that accesses the content of `solution.code`. Since the top-level block is a `lastOf`, the output of the program is the result of this last block. The left column of Table 1 shows some possible outputs of the program, since LLM responses are not deterministic.

**Probabilistic prompt programming.** PPDL extends PDL with a single additional primitive, `factor`, but fundamentally changes the execution model by turning the program into a distribution over traces. In PPDL, the user can use `factor` to score executions. The program is then bootstrapped into a probabilistic framework that uses those scores in various algorithms. The output of PPDL is a distribution of results, from which the user can sample the

relies on Monte Carlo approximate inference algorithms that launch a series of independent executions, called *particles*, to estimate the posterior distribution. For example, using 5 particles, we obtain the distribution of solutions shown in Table 1. The solution with high probability is correct, while the other two are incorrect. Three of the five particles found the first solution, so majority voting would have yielded a wrong solution. Appendix A shows some step-by-step executions of the program.

The advantage of constraints is that they attach probabilities to each response, allowing PPDL to return a correct response even if that response is not part of the majority. In this example, we used importance sampling, but we could have used the same program and selected majority voting or Sequential Monte Carlo. PPDL allows the user to write the logic of the flow once, and experiment with different inference scaling approaches without needing additional code. Appendix B presents alternative implementations of Figures 2 and 3 in other frameworks to illustrate the refactoring required when switching inference scaling algorithms.

Inference scaling approaches depend on *constraints* to steer execution traces and select solutions. Constraints can be hard (e.g., failing tests) or soft (e.g., being scored on a scale), cheap (e.g., checking syntax) or expensive (e.g., calling a large model). Sometimes constraints are sound, sometimes complete, but even when neither sound nor complete, they can improve predictive performance. Some inference scaling literature (Cobbe et al., 2021; Stroebl et al., 2026) refers to the source of constraints as "verifiers," but we avoid that terminology because it misleadingly implies soundness. Other literature, inspired by reinforcement learning, refers to constraints as *rewards* and distinguishes process rewards (in the middle of an execution trace) from outcome rewards (at the end of a completed trace). Constraints can come from models, which can be either fine-tuned for this task (e.g., process reward models, PRMs) or prompted for this purpose (Zheng et al., 2023). PPDL supports all of the above forms of constraints via the probabilistic `factor` as a single orthogonal language construct.

## 3. Language and Approach

This section presents how PPDL programs execute, formalizing their behavior through a probabilistic semantics.

**Overview.** As shown in Figure 1, PPDL lets users express an LLM-based flow as a program with factors (top left), then explores traces of that program to obtain a distribution of output values (bottom left). This section formalizes this intuition via a semantics of the form $p/S \rightsquigarrow \lambda v.\Pr(v)$, where $p$ is a program, $S$ is its environment state, $\rightsquigarrow$ means reduces to, and $\lambda v.\Pr(v)$ denotes a categorical probability distribution over output values $v$.

To define such a semantics over all traces, we first define a semantics for a single trace, which has the form $p/S \Rightarrow^* v/S'$. We refer to the single-trace semantics as a *weighted sampler*, where the weights come from factors and the sampling happens via LLM model calls. The weighted sampler is defined using reduction rules of the form $p/S \Rightarrow p\_or\_v/S'$. Each reduction rule transforms its left-hand side (program $p$ and environment $S$) into its right-hand side (the remainder of the program at the next factor or a final value $v$, with updated environment $S'$). The reduction $\Rightarrow^*$ is the transitive closure of $\Rightarrow$ and continues until it produces a value.

During execution, the environment $S$ keeps track of two special variables: $S[\texttt{pdl\_context}]$ accumulates messages, and $S[\texttt{pdl\_score}]$ is the trace likelihood. LLM calls introduce randomness: each model call samples a response from the distribution defined by an LLM given the current `pdl_context`, resulting in a new message appended to `pdl_context`. Each `factor` increments `pdl_score` by a value provided as a parameter. A hard constraint corresponds to a `factor` with parameter $-\infty$ (probability 0 in log space). As in classic probabilistic programming languages (Bingham et al., 2019; Carpenter et al., 2017), we can assume that a value was sampled from a distribution using a `factor` whose parameter is the log-density of the distribution at that point. The value of `pdl_score` thus represents how well the trace satisfies the user constraints.

Section 3.1 formalizes the weighted sampler by defining the syntax and semantics of core PPDL blocks (see Appendix C for the remaining blocks). The ideal semantics ($\rightsquigarrow$) uses an infinite weighted sum over all possible traces ($\Rightarrow^*$). Section 3.2 describes the probabilistic inference engines, each of which computes a finite approximation of the ideal semantics. Both the ideal semantics and the approximate inference engines rely on the weighted sampler. Section 3.3 describes the parallelized implementation of the weighted sampler and the inference engines.

### 3.1. Weighted Sampler

Figure 4 presents the syntax of a simplified kernel of PPDL, showing its context-free grammar in Backus-Naur Form. A program is a block comprising a set of variable definitions, the block body (i.e., the instruction to evaluate), a parser to extract the result from the return value of the body, a flag indicating whether the result contributes to the context, and the name of the variable in which to store this result. The body of a block can be an expression (`data`), code for a tool call, an LLM call with its input, a conditional, a sequence of blocks with a join operator to gather result values, a loop with a join operator to gather the results of each iteration, a function definition, a function call, or a factor statement to update the score. The examples from

$$
\begin{aligned}
pdl \quad &::= \{ \, \texttt{defs:} \{x\!:\!pdl, \ldots, x\!:\!pdl\}, \\
&\qquad body, \\
&\qquad \texttt{parser:} parser, \\
&\qquad \texttt{contribute:} contribute, \\
&\qquad \texttt{def:} x \qquad\qquad\qquad\qquad \} \\
body \quad &::= \texttt{data:} expr \\
&\quad | \; \texttt{code:} pdl, \texttt{lang:} string \\
&\quad | \; \texttt{model:} expr, \texttt{input:} pdl \\
&\quad | \; \texttt{if:} expr, \texttt{then:} pdl, \texttt{else:} pdl \\
&\quad | \; \texttt{sequence:} [pdl, \ldots, pdl], \texttt{join:} join \\
&\quad | \; \texttt{while:} expr, \texttt{repeat:} pdl, \texttt{join:} join \\
&\quad | \; \texttt{function:} types, \texttt{return:} pdl \\
&\quad | \; \texttt{call:} expr, \texttt{args:} expr \\
&\quad | \; \texttt{factor:} expr \\
expr \quad &::= \texttt{null} \mid bool \mid number \mid string \mid \$\{jinja\_expr\} \\
&\quad | \; [expr, \ldots, expr] \mid \{x\!:\!expr, \ldots, x\!:\!expr\}
\end{aligned}
$$

*Figure 4.* Syntax of PPDL (using the flow-style of YAML).

Section 2 can be compiled into this kernel. For instance, blocks without an explicit `defs` clause can be compiled to use an empty `defs`, and a `model` block without an `input` clause can be compiled to use `pdl_context` as input.

The top rule of Figure 5 is the main rule defining the weighted sampler reduction $p/S \Rightarrow p\_or\_v/S'$. The notation for the rules in this paper places premises (sub-steps) above a horizontal line and conclusions (the reduction being defined) below the line. The premises for evaluating a block begin by evaluating the variable definitions, then running the block body to produce a value $v$, parsing $v$ into a new value $v'$, and finally contributing $v'$ to the background context and adding it to the environment. Contribution to the background context is performed by adding a message to the sequence in `pdl_context` (we omit the role in the semantics for simplicity):

$$
\frac{\begin{array}{c} ctx = S[\texttt{pdl\_context}] + [\{\texttt{content:} v\}] \\ S' = S[\texttt{pdl\_context} \leftarrow ctx] \end{array}}{S, v, [\texttt{pdl\_context}] \xrightarrow{contrib} S'}
$$

The rest of Figure 5 shows the key block-body reduction rules. The `model` block evaluates the model name and input expressions to get an actual model name $m$ and a sequence of messages $i$, then samples the LLM corresponding to $m$ conditioned on $i$ to obtain the output. The `factor` block computes a score $w$ and increments the `pdl_score` variable by this value. It rewrites into a block that returns the empty string. The rule for a function `call` illustrates the handling of the special variables `pdl_context` and `pdl_score`. As in functional programming languages, it first retrieves the function definition along with its local environment, evaluates the arguments, checks that they are well-typed, then evaluates the function body in the local environment to compute the returned value. A key property of

PPDL is that `pdl_context` and `pdl_score` are propagated through function calls, allowing accumulation to continue.

Now that we have a weighted sampler, we can define the ideal semantics of a PPDL program as an infinite number of executions of the sampler ($N \to \infty$) and normalize the weights to obtain a categorical distribution:

$$
\frac{\begin{array}{c} \{p/S \Rightarrow^* v_i/S_i\}_{1 \leq i \leq N} \\ \{w_i = \exp(S'_i[\texttt{pdl\_score}])\}_{1 \leq i \leq N} \quad W = \sum_{1 \leq i \leq N} w_i \end{array}}{p/S \rightsquigarrow \lambda U. \sum_{1 \leq i \leq N}(w_i/W) \times \delta_{v_i}(U)}
$$

The premise $\{p/S \Rightarrow^* v_i/S_i\}_{1 \leq i \leq N}$ denotes a collection of $N$ executions of the program $p$ starting from the same initial environment $S$. Each execution reduces $p$ to a final value $v_i$ and environment $S_i$. Each $S_i$ contains the variable `pdl_score`, which stores the unnormalized weight (in log space) accumulated along the corresponding execution trace. The sum $W$ normalizes the score of each result $v_i$. The final result is a categorical distribution that, given a set of values $U$, returns its probability (where $\delta_{v_i}(U)$ is the Dirac distribution that returns 1 when $v_i \in U$).

### 3.2. Inference Engines

The ideal semantics are intractable as $N \to \infty$, but probabilistic programming inference algorithms provide ways to approximate them. For example, if $N$ is bounded, the rule defining the semantics corresponds to the *Importance Sampling (IS)* algorithm. Each reduction $p/S \Rightarrow^* v_i/S_i$ is a *particle* computing a value $v_i$ with its score $S_i[\texttt{pdl\_score}]$. Similarly, majority voting uses the same rule with uniform weights, ignoring $S_i[\texttt{pdl\_score}]$.

We can also use our weighted sampler to describe the *Sequential Monte Carlo (SMC)* algorithm (Del Moral et al., 2006). With SMC, during execution, particles with a low score drop their current execution path and restart from the state of a particle with a higher score. The state selection is made by sampling from the distribution of program states at each *resampling point*. In PPDL, the resampling points correspond to the `factor` blocks (i.e., when the reduction $\Rightarrow$ returns a program block $p$ instead of a value $v$).

The SMC algorithm is formalized as follows:

$$
\frac{\delta_{p,S} \rightsquigarrow^* D_{v,S'}}{p/S \rightsquigarrow \pi_1(D_{v,S'})}
$$

$$
\frac{\begin{array}{c} \{p_i/S_i \Rightarrow p'_i/S'_i \mid (p_i, S_i) = sample(D_{p,S})\}_{1 \leq i \leq N} \\ \{w_i = \exp(S'_i[\texttt{pdl\_score}])\}_{1 \leq i \leq N} \quad W = \sum_{1 \leq i \leq N} w_i \end{array}}{D_{p,S} \rightsquigarrow \lambda U. \sum_{1 \leq i \leq N}(w_i/W) \times \delta_{p'_i, S'_i[\texttt{pdl\_score} \leftarrow 0]}(U)}
$$

Starting from a program $p$ and environment $S$, we turn this initial state into a Dirac distribution ($\delta_{p,S}$) and then apply the reduction $D_{p,S} \rightsquigarrow D_{p',S'}$, which rewrites a distribution

$$\frac{defs/S \xrightarrow{defs} \{\}/S_1 \quad block\_body/S_1 \to v/S_2 \quad parser, v \xrightarrow{parse} v' \quad S_2, v', contribute \xrightarrow{contrib} S_3 \quad S' = S_3[x \leftarrow v']}{\{\texttt{defs}:defs, block\_body, \texttt{parser}:parser, \texttt{contribute}:contribute, \texttt{def}:x\}/S \Rightarrow v'/S'}$$

$$\frac{S \vdash model \Downarrow m \quad S \vdash input \Downarrow i \quad sample(m, i) = v}{\texttt{model}:model, \texttt{input}:input/S \to v/S} \qquad \frac{S \vdash e \Downarrow w \quad S' = S[\texttt{pdl\_score} \leftarrow S[\texttt{pdl\_score}] + w]}{\texttt{factor}:e/S \to \texttt{data}:\texttt{""}/S'}$$

$$\frac{S \vdash f \Downarrow \texttt{function}:t, \texttt{return}:p, \texttt{scope}:S_f \quad S \vdash args \Downarrow args' \quad args' \in t}{p/S_f[\texttt{pdl\_context} \leftarrow S[\texttt{pdl\_context}], \texttt{pdl\_score} \leftarrow S[\texttt{pdl\_score}]] + args' \Rightarrow v/S'_f}{S' = S[\texttt{pdl\_context} \leftarrow S'_f[\texttt{pdl\_context}], \texttt{pdl\_score} \leftarrow S'_f[\texttt{pdl\_score}]]}{\texttt{call}:f, \texttt{args}:args/S \to v/S'}$$

*Figure 5.* Key reduction rules for blocks ($p/S \Rightarrow p\_or\_v/S'$) and block-bodies ($b/S \to b\_or\_v/S'$). The reduction $S \vdash e \Downarrow v$ represents the evaluation of an expression $e$ into a value $v$ in the environment $S$. The full semantics is provided in Appendix C.

of program states (pairs of program and environment) into a new distribution. When all programs in the distribution reduce to values ($D_{v,S'}$), the result is the pushforward of the distribution across the first projection ($\pi_1$), yielding the distribution of return values alone.

The reduction $D_{p,S} \rightsquigarrow D_{p',S'}$ samples $N$ program/environment pairs from $D_{p,S}$ and applies the weighted sampler $\Rightarrow$ to compute the next distribution.

### 3.3. Parallel Execution

PPDL is an interpreted language. Its implementation directly follows the semantics. Both the implementation of the inference engine and of the weighted sampler are parallelized.

Bayesian inference algorithms such as IS and SMC are inherently parallel, since they rely on independent executions of particles. With IS, full traces can be executed in parallel. SMC introduces a synchronization point at each resampling step, when the distribution of intermediate program states is built and sampled. The PPDL interpreter implements this parallelism using multi-threading.

Within the execution of a particle, the PPDL interpreter parallelizes independent model calls using futures (Halstead, 1985). Model calls return a promise of a response rather than blocking. When an expression is evaluated, the interpreter traverses the relevant data structures to await the responses of only those model calls on which the expression depends. To limit dependencies, the data structures created by PPDL, especially `pdl_context`, are lazy (Okasaki, 1998), ensuring that only the accessed cell of an array or field of an object is awaited.

## 4. Evaluation

### 4.1. PPDL as an Inference Scaling Framework

This section evaluates PPDL as a unified inference-scaling framework across multiple tasks and models. Our goal is not to claim state-of-the-art task performance, but to demonstrate that PPDL enables users to write a single program and experiment with different inference algorithms without rewriting the flow logic. This allows systematic exploration of which inference strategy works best for a given task and model combination.

**Models.** We used granite4-small, granite4-micro, gpt-oss-120b, gpt-oss-20b, llama4-maverick, and llama4-scout in our experiments, covering different families and sizes of models. Appendix D includes their release and cutoff dates. We set the model temperature for all experiments to 0.8, and use the default reasoning effort for gpt-oss-120b (medium). All other hyperparameters were left at their default values.

**Benchmarks.** We use benchmarks that span a variety of tasks and cap all datasets at 500 samples:

- **Reasoning:** GSM8k (Cobbe et al., 2021) is a dataset of grade-school level math problems; Math500 (Math-500 Dataset) is a curated selection of 500 problems from the Math dataset (Hendrycks et al., 2021), spanning algebra, geometry, number theory, pre-calculus, and probability.
- **Code generation:** MBPP (Austin et al., 2021) is a collection of mostly basic Python programming tasks; LiveCodeBench (Jain et al., 2025) is a dynamic contamination-free benchmark using fresh problems from LeetCode, AtCoder, and Codeforces. We selected the code generation task and its latest version (v6) with a time window from 8/1/2024 to 5/1/2025.
- **Question answering:** Fever (Thorne et al., 2018) is a fact checking dataset.

*Table 2.* Accuracy comparison across benchmarks, models, and inference algorithms. IS@1 is the baseline (single execution), Maj. Voting (Majority Voting), IS (Importance Sampling), and SMC (Sequential Monte Carlo) use 5 particles each, and pass@k represents the oracle upper bound. Bold indicates best performance among inference algorithms. All results show mean ± standard deviation over 3 runs.

| model | IS@1 | Maj. Voting | IS | SMC | pass@k |
|---|---|---|---|---|---|
| **GSM8k** | | | | | |
| granite4-small | 83.8% ± 0.3 | 90.4% ± 0.9 | **93.7%** ± 0.5 | 92.3% ± 0.3 | 96.1% ± 0.3 |
| granite4-micro | 79.5% ± 0.9 | 87.1% ± 1.6 | **87.8%** ± 0.4 | 84.2% ± 0.6 | 93.7% ± 0.1 |
| llama4-scout | 93.9% ± 0.2 | 94.5% ± 0.3 | 94.4% ± 0.3 | **94.6%** ± 0.3 | 95.5% ± 0.2 |
| llama4-maverick | 95.3% ± 0.2 | **95.7%** ± 0.4 | 95.4% ± 0.3 | 95.3% ± 0.3 | 96.7% ± 0.3 |
| gpt-oss-120b | 86.9% ± 0.3 | 90.4% ± 0.8 | **92.2%** ± 0.3 | 90.8% ± 0.0 | 95.1% ± 0.1 |
| gpt-oss-20b | 90.2% ± 1.2 | **93.1%** ± 0.5 | 92.7% ± 0.4 | 91.9% ± 0.8 | 95.5% ± 0.2 |
| **Math500** | | | | | |
| granite4-small | 60.7% ± 0.4 | 62.1% ± 0.6 | 64.0% ± 1.1 | **65.5%** ± 1.7 | 75.7% ± 0.3 |
| granite4-micro | 47.4% ± 0.6 | 47.7% ± 1.2 | **52.5%** ± 1.1 | 51.1% ± 0.7 | 66.5% ± 0.5 |
| llama4-scout | 66.3% ± 0.1 | 65.7% ± 0.5 | 67.3% ± 0.4 | **67.5%** ± 0.6 | 71.9% ± 0.3 |
| llama4-maverick | 69.9% ± 0.3 | 69.7% ± 0.6 | **71.5%** ± 0.9 | 70.1% ± 0.5 | 74.7% ± 0.6 |
| gpt-oss-120b | 74.2% ± 0.2 | 73.8% ± 1.1 | **74.8%** ± 1.0 | 74.5% ± 0.3 | 77.9% ± 0.3 |
| gpt-oss-20b | 74.1% ± 0.1 | 74.0% ± 1.1 | **75.7%** ± 0.5 | 74.9% ± 0.4 | 79.0% ± 0.2 |
| **MBPP** | | | | | |
| granite4-small | 69.7% ± 1.9 | 69.7% ± 2.9 | **80.5%** ± 2.1 | 80.2% ± 2.0 | 92.0% ± 1.2 |
| granite4-micro | 71.9% ± 1.0 | 72.9% ± 2.5 | **75.1%** ± 2.6 | 74.8% ± 0.3 | 89.5% ± 1.8 |
| llama4-scout | 82.5% ± 0.9 | 81.7% ± 2.0 | **86.1%** ± 2.1 | 85.4% ± 1.6 | 92.0% ± 1.6 |
| llama4-maverick | 90.3% ± 0.5 | 90.4% ± 2.3 | **91.4%** ± 1.0 | 90.9% ± 1.5 | 95.9% ± 0.9 |
| gpt-oss-120b | 92.4% ± 0.2 | 93.6% ± 0.3 | **94.1%** ± 1.1 | 93.4% ± 1.0 | 97.8% ± 0.3 |
| gpt-oss-20b | 92.3% ± 0.9 | **93.1%** ± 1.8 | 91.7% ± 1.2 | 92.4% ± 1.0 | 98.0% ± 0.5 |
| **LiveCodeBench** | | | | | |
| granite4-small | 15.8% ± 0.3 | 15.5% ± 0.8 | **17.4%** ± 1.1 | 15.7% ± 0.6 | 24.0% ± 0.7 |
| granite4-micro | 11.1% ± 0.3 | **11.5%** ± 1.3 | 10.7% ± 0.6 | 11.1% ± 0.9 | 18.2% ± 0.7 |
| llama4-scout | 23.9% ± 0.3 | 24.5% ± 0.6 | **24.9%** ± 0.4 | 23.9% ± 1.0 | 30.3% ± 0.6 |
| llama4-maverick | 29.0% ± 0.2 | 28.5% ± 0.3 | 29.1% ± 0.2 | **29.3%** ± 0.1 | 36.7% ± 0.2 |
| gpt-oss-120b | 30.3% ± 0.8 | 19.5% ± 1.5 | **36.1%** ± 2.0 | 35.8% ± 0.8 | 48.2% ± 0.3 |
| gpt-oss-20b | 20.8% ± 0.5 | 12.5% ± 1.2 | 8.0% ± 0.9 | **15.9%** ± 0.8 | 40.2% ± 0.3 |
| **Fever** | | | | | |
| granite4-small | 76.6% ± 0.6 | 77.5% ± 0.8 | 77.9% ± 1.0 | **78.1%** ± 1.3 | 88.3% ± 0.9 |
| granite4-micro | 71.7% ± 0.2 | 73.6% ± 0.3 | **73.7%** ± 0.8 | 73.3% ± 1.2 | 90.3% ± 1.1 |
| llama4-scout | 85.6% ± 0.4 | 85.0% ± 0.2 | **85.9%** ± 1.5 | 84.1% ± 0.1 | 93.7% ± 0.8 |
| llama4-maverick | 81.6% ± 0.3 | **81.7%** ± 0.6 | 80.9% ± 0.4 | 81.0% ± 0.2 | 85.7% ± 0.3 |
| gpt-oss-120b | 86.5% ± 0.2 | **87.5%** ± 0.5 | 86.7% ± 0.3 | 87.2% ± 0.0 | 92.7% ± 0.6 |
| gpt-oss-20b | 79.9% ± 0.4 | **81.3%** ± 0.2 | 79.9% ± 1.0 | 81.2% ± 0.5 | 90.6% ± 0.8 |

**Inference scaling algorithms.** We consider the following inference scaling algorithms: majority voting (Maj. Voting), importance sampling (IS), and sequential Monte Carlo (SMC). We run all experiments with 5 particles.

**PPDL Programs.** For each benchmark, we write a PPDL program to capture the logic of interacting with LLMs and add factors to shift the distribution toward desirable results using both LLM-based judges and rule-based ones. The complete PPDL implementations are in Section E.

- **Reasoning:** The program asks the LLM to solve the math problem, then extracts the answer and asks the same LLM to judge the correctness of the solution and score it according to the formula shown in Section 2.
- **Code generation:** The program first asks the LLM to generate an English plan for the problem at hand, then

uses an LLM to judge the validity of the plan (scoring accordingly). It then generates a Python function corresponding to the plan by calling the model. It extracts the Python code and uses rule-based judges to check that the solution contains function definitions and to evaluate the number of warnings using flake8. Finally, the same LLM is used to judge the correctness of the solution.

- **Question answering:** The program first asks the question to the LLM, enabling it to use a Wikipedia search tool. This may result in tool-use requests. We then check whether the topic(s) to be searched are sensible using the same LLM as a judge and call the search tool. We use LLM-as-judge to check whether enough evidence has been gathered and finally call the LLM again to verify or refute the original question given the gathered evidence.

**Results.** Table 2 reports the results of our experiments. We perform three runs for each task, model, and algorithm and report the mean accuracy and standard deviation. The column labeled IS@1 is the baseline corresponding to a program executed without inference scaling. Similarly, pass@k is an oracle upper bound, where at least one out of $k$ particles has the correct answer. These benchmarks demonstrate how easily a user can experiment with different inference scaling algorithms for a given task and model and choose between them without writing additional code. This is particularly interesting for SMC, which has a more intricate implementation. The results show the accuracy improvements achieved by inference scaling over the baseline and that while simple majority voting is the best-performing algorithm in some cases (e.g., GSM8k with gpt-oss-20b), in other cases importance sampling or SMC performs best.

In general, importance sampling (IS) and SMC are preferable to majority voting when the `factor` statements provide meaningful information, as they use the factors to shift probability mass toward better traces. Between IS and SMC, SMC is particularly advantageous for longer, deeper flows with informative intermediate factors. IS only reweights completed traces, while SMC can resample during execution at factor points, allowing computation to be redirected toward more promising partial traces earlier in the flow. This gives SMC a more meaningful exploration of the search space in multi-step settings, as demonstrated in the theorem-proving case study below.

### 4.2. Case Study: Theorem Proving in Rocq

As a case study, we use PPDL to implement a theorem proving agent for Rocq, formerly known as Coq (Rocq Prover Team), a proof assistant, similar to Lean or Isabelle, that can automatically check mathematical proofs. The complete PPDL implementation is provided in Section E.6.

We focus on the MiniF2F dataset (Zheng et al., 2021; Jiang et al., 2022), a popular benchmark for evaluating theorem proving agents (Polu & Sutskever, 2020; Thakur et al., 2024; Mikuła et al., 2023; Wang et al., 2024). The test split comprises 244 high-school level math exercises, ranging from simple algebra to International Mathematics Olympiad problems. While the Lean version of this benchmark is already saturated, with state-of-the-art specialized models with aggressive inference scaling reaching up to 99% accuracy (Varambally et al., 2025), the Rocq version (Viennot et al., 2025) is more recent, and the contamination risk is minimal. The goal of this experiment is not to compare our agent with state-of-the-art solutions, but to measure how much probabilistic inference can improve the performance of a base model on a challenging task.

**Experimental setup.** We adapt an experiment originally designed for Lean to the Rocq prover to study how different prompting and inference scaling strategies affect LLM-based theorem proving (G. Narozniak, personal communication, 2026). A *full proof* generation agent alternates between two steps: LLM-based proof synthesis and prover-based verification. The agent implements a simple proof/repair loop: if the verification step fails, the agent lists all errors and prompts the LLM to fix the proof.

For each exercise, we compare three inference scaling strategies under a limit on the total number of tokens that can be used to find the proof: 1) IS@1: repeatedly retry until a proof is found or the token budget is exhausted (this corresponds to sequential scaling); 2) IS@$k$: run $k$ particles and stop as soon as one of them finds a proof or the token budget is exhausted (this is equivalent to pass@k thanks to the perfect verifier and because scores are ignored when a proof is found); 3) SMC@$k$: similar to IS, but use the number of errors at each attempt to score the particles and prioritize the most promising proofs. When a proof is found, the agent terminates the execution of all parallel particles and returns the successful proof.

Using PPDL, we evaluate these three strategies on the same program by adjusting the runtime configuration, varying only the inference algorithm and particle count. We run all experiments with gpt-oss-120b, a temperature of 1.0, and a budget of 1M tokens.

**Results.** Table 3 presents a summary of the results. We observe that with a constant number of tokens the accuracy varies from 30% (72.7/244 for IS@1) to 39% (95/244 for SMC@40). With a fixed number of particles, SMC always performs better than IS. SMC can resample during execution at factor points, allowing computation to be redirected toward more promising partial traces earlier in the flow. This gives SMC a more meaningful exploration of the search space in multi-step settings. SMC also benefits from

*Table 3.* Number of problems solved out of the 244 MiniF2F-Rocq problems. We present the average ± one standard deviation over three runs.

| Particles | IS | SMC | delta |
|---|---|---|---|
| 1 | 72.7 ± 2.9 | - | - |
| 5 | 82.7 ± 3.9 | 84.7 ± 1.2 | +2.0 |
| 10 | 84.3 ± 3.9 | 91.3 ± 1.7 | +7.0 |
| 20 | 87.0 ± 1.6 | 94.3 ± 2.1 | +7.3 |
| 40 | 87.7 ± 2.4 | 95.0 ± 2.4 | +7.3 |

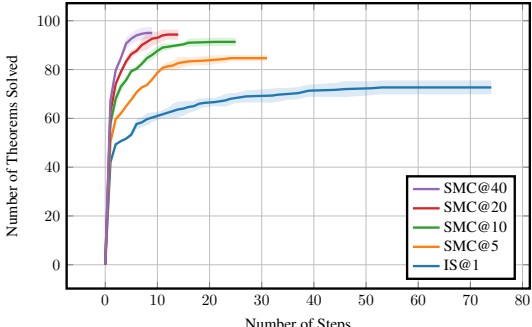

*Figure 6.* Number of MiniF2F-Rocq problems solved as a function of proof/repair steps. Each curve shows the mean over three runs, with shaded regions indicating one standard deviation.

having more particles since larger particle populations help preserve diversity during resampling and reduce the risk of collapsing too early onto a narrow set of traces.

Since the number of tokens is constant, there is a trade-off between the number of proof/repair steps and multiple particles searching solutions in parallel. Figure 6 shows this trade-off for the SMC algorithm (results are similar for IS and shown in Figure 26 in the appendix). A single particle (IS@1) leverages prover feedback across up to 74 sequential repairs whereas SMC@40 goes down to 9 steps. Parallel exploration is more efficient for exploring the proof space and can prove more theorems in a single step, but it exhausts the token budget more quickly.

Our agent is relatively simple, but optimizing this exploration/exploitation trade-off could be even more beneficial for more advanced prompting strategies that leverage feedback from multiple tools (e.g., the prover, specialized search engines, or a natural language reasoner).

## 5. Related Work

There is not much prior work at the intersection between prompt programming and probabilistic programming. LLaMPPL (Lew et al., 2023) and GenML Control (Loula et al., 2025) apply probabilistic inference engines to the decoding of LLM-generated tokens with SMC and both soft and hard constraints. But unlike our work,

they focus on a single LLM call, not a flow of LLM and tool calls. LMQL features hard constraints and beam search over traces (Beurer-Kellner et al., 2023); unlike our work, it lacks soft constraints and probabilistic inference such as SMC. Similarly, EnCompass also supports beam search over traces (Li et al., 2025); however, as it does not track how uncertainty accumulates across factors, it also lacks probabilistic inference. Rollout Roulette uses a process reward model with SMC over token sequences (Puri et al., 2025); in contrast, our work supports a more general set of soft and hard constraints, and supports multi-call flows. Bertsch et al. (2023) reframe inference scaling as probabilistic inference to find the Minimum Bayes Risk sample, but do not explore SMC nor LLM-and-tool flows.

We take inspiration from the rich literature on probabilistic languages (PPLs) that predates prompt programming with LLMs. An overview paper by Gordon et al. (2014) introduces core constructs of probabilistic programming and inference via the Prob PPL. The Stan PPL has been widely adopted in practice (Carpenter et al., 2017). The Church PPL is based on lambda calculus (Goodman et al., 2008). Like PPDL, each of Prob, Stan, and Church tracks and updates a log-probability along a trajectory; but unlike PPDL, none of them interface with LLMs. Some more recent PPLs do interface with neural networks, such as Pyro (Bingham et al., 2019) and DeepStan (Baudart et al., 2021a), but they do not focus on language models, let alone prompt programming.

Another source of inspiration for PPDL is the growing literature on prompt programming languages. We already discussed LMQL and PDL. Guidance focuses on hard constraints to guide generation (Lundberg et al., 2022). Both DSPy (Khattab et al., 2024) and AutoPDL (Spiess et al., 2025) provide support for optimizing prompt programs for an objective function, which could be viewed as a final soft constraint. Other prompting languages include SGLang (Zheng et al., 2024), Vieira (Li et al., 2024), APPL (Dong et al., 2025), and Opp (Mell et al., 2025). Unlike PPDL, none of these are probabilistic languages; while several of the papers include inference scaling examples, those are written by hand, not orthogonal like in PPDL.

Parallel inference scaling scales the number of independent attempts for solving a problem (where each attempt can be a full trace). Examples include Self-consistency (Wang et al., 2023) and Large Language Monkeys (Brown et al., 2024). PPDL makes parallel inference scaling implicit and orthogonal. Sequential inference scaling scales the number of tokens in a single LLM call or the number of LLM calls in a single trajectory. Examples include chain-of-thought (Wei et al., 2022), Self-refine (Madaan et al., 2023), and ReAct agents (Yao et al., 2023b). PDL has been designed to facilitate the implementation of flows that

embody these sequential inference scaling approaches, and PPDL directly inherits this property. Besides parallel and sequential, a third style is search-based inference scaling, e.g., Yao et al. (2023a) and Zhou et al. (2024). PPDL makes search-based inference scaling implicit and orthogonal.

Rooted in earlier work on automated machine learning (Feurer et al., 2015; Baudart et al., 2021b), automated prompt optimization uses known question/answer pairs to improve prompts for future questions. More recently, *instance optimization* applies prompt optimization for a specific new question; examples include TextGrad (Yuksekgonul et al., 2025), Trace (Cheng et al., 2024), and Gepa (Agrawal et al., 2025). Unlike our work, none are PPLs or use probabilistic inference engines. In future work, we plan to integrate some of these techniques.

# 6. Conclusion

This paper introduces PPDL, a new programming language that unifies prompt programming and probabilistic programming. Prompt programming languages make LLM prompting easy, but struggle with inaccuracy and uncertainty. One solution for boosting accuracy is inference scaling, but unfortunately, manually adding inference scaling to a prompt program sacrifices simplicity and flexibility. Instead, PPDL lets users write the core logic in a simple prompting language, and the interpreter applies orthogonal probabilistic inference engines to search the space of trajectories. As a bonus, since the program returns a distribution, users also get a better feeling for the uncertainty of results.

**Limitations.** While useful, inference scaling is no silver bullet (Stroebl et al., 2026). Any steering based on factors can only be as good as the information in those factors, which is often imperfect, e.g., based on an LLM-as-a-judge (Zheng et al., 2023). In future work, we plan to mitigate that limitation by calibrating factors, so multiple constraints carry the appropriate relative weight.

# Acknowledgements

We would like to thank Theo Stoskopf for all the help and support with the rocq-ml-toolbox (https://github.com/LLM4Rocq/rocq-ml-toolbox) that we used for the theorem-proving case study.

# Impact Statement

This paper presents work whose goal is to advance the field of Machine Learning. There are many potential societal consequences of our work, none of which we feel must be specifically highlighted here.

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

## A. Example Walk-Through

This section illustrates the execution of the PPDL program from Figure 3 using two different inference algorithms: Importance Sampling (IS) and Sequential Monte Carlo (SMC). Both examples use 5 particles to solve the centered hexagonal number problem, where the task is to generate a Python function that computes the $n$th centered hexagonal number using the formula $1 + 3n(n - 1)$.

**Importance Sampling execution (Table 4).** In IS, all particles execute independently from start to finish. Each particle generates a plan (first row), which is then scored using an LLM-as-a-judge constraint (second row). The particles then generate code solutions (third row), which are scored using a linter-based constraint (fourth row).

The key insight from this execution is how the scoring mechanism differentiates solution quality. Particles 0 and 4 generate correct solutions with the formula `1 + 3*n*(n-1)`, and receive very low penalty scores (approximately $-0.69$ after both scoring steps). Particles 1, 2, and 3 generate incorrect formulas and receive much higher penalty scores (ranging from $-14.25$ to $-17.44$).

When these scores are normalized into probabilities, particles 0 and 4 each have approximately 50% probability, while the incorrect solutions have negligible probabilities (less than $0.0004\%$ combined). This demonstrates how IS uses scoring to amplify the probability of correct solutions. Without scoring (i.e., using majority voting), the probability of sampling a correct solution would be only 0.4, since only 2 out of 5 particles found the correct formula.

**Sequential Monte Carlo execution (Table 5).** SMC differs from IS by introducing a resampling step after the first scoring point. Initially, all 5 particles generate plans and receive scores (first two rows). Particles 1 and 2 have significantly better scores ($-0.000\,000\,002$ and $-0.000\,000\,009$) because their plans contain the correct formula, while the other particles have much worse scores.

The resampling step (indicated by the arrows in Table 5, similar to Figure 1 from the introduction) duplicates the promising particles 1 and 2 while discarding the poorly-scoring particles 0, 3, and 4. After resampling, the particle population consists of three copies derived from particle 1 and two copies derived from particle 2. These resampled particles then continue execution to generate code solutions (third row).

Crucially, all five particles in the second phase now generate correct solutions, because they all descended from particles with good plans. After the final scoring step, all particles have identical scores ($-0.69$), resulting in a uniform distribution where each solution has equal probability.

```
1  llm = "watsonx/meta-llama/llama-4-maverick-..."
2  problem_statement = (
3    "Write a function to find nth centered hexagonal number.\n"
4    "assert centered_hexagonal_number(10) == 271\n"
5  )
6
7  messages = [
8    {
9      "content": (
10       "Generate an English plan for how to generate code "
11       f"for the following problem: { problem_statement }"
12     ),
13     "role": "user",
14   }
15 ]
16 response = completion(model=llm, messages=messages)
17 plan = response.choices[0].message
18 messages = messages + [ plan ]
19 messages = messages + [
20   {
21     "content": (
22       "Generate a complete executable Python function "
23       "definition corresponding to the above plan and "
24       "problem. Generate only a single function definition."
25     ),
26     "role": "user",
27   }
28 ]
29 response = completion(model=llm, messages=messages)
30 solution_str = response.choices[0].message["content"]
31 solution_match = re.fullmatch(
32   r"(.|\n)*```python\n(?P(.|\n)*?)```(.|\n)*",
33   solution_str, flags=re.M
34 )
35 solution = solution_match.group("code")
```

*Figure 7.* Implementation in Python using LiteLLM of Figure 2.

This execution demonstrates SMC's ability to focus computational resources on promising execution paths early in the computation, leading to higher-quality final results.

The contrast between these two executions highlights the fundamental difference between IS and SMC: IS evaluates all particles independently and uses scoring only at the end to weigh results, while SMC actively steers the particle population toward promising regions of the solution space through intermediate resampling steps.

## B. Possible alternatives for Figures 2 and 3

This section demonstrates how the PPDL programs from Figures 2 and 3 can be implemented using standard Python libraries. These implementations illustrate both the expressiveness advantages of PPDL and its orthogonality between program and inference scaling startegy. We present four implementations: a baseline using LiteLLM, importance sampling (IS) and sequential Monte Carlo (SMC) variants with LiteLLM, and an SMC implementation using LangChain that exploits parallelism.

First, Figure 7 corresponds to the implementation of Figure 2 using LiteLLM (https://github.com/BerriAI/litellm) to

*Table 4.* Example of execution of the program provided in Figure 3 using Importance Sampling. The columns represent the state of the different particles, and the rows represent the values of the variables through the execution. The circles visualize probability magnitudes as in Figure 1. There is a probability of 0.999 999 453 to sample a correct solution (particles 0 and 4) in this distribution. If we ignore the scores, this trace corresponds to a majority voting execution. In this case, the probability of sampling a correct solution is 0.4.

| Variable | Particle 0 | Particle 1 | Particle 2 | Particle 3 | Particle 4 |
|---|---|---|---|---|---|
| `plan` | ... - The formula `3*n*(n+1) + 1` is a more direct representation of the nth centered hexagonal number. ... | ... The formula to calculate the nth centered hexagonal number is given by: $1 + 6 * (1 + 2 + ... + n)$, which simplifies to $1 + 6 * (n * (n + 1)/2)$. ... | ... The formula for the nth centered hexagonal number is given by: $1 + 6 * (1 + 2 + ... + n)$, which simplifies to $1 + 6 * (n * (n + 1) / 2)$. ... | ... The formula for the nth centered hexagonal number is given by: $1 + 6 * (1 + 2 + ... + n)$, which simplifies to $1 + 6 * (n * (n + 1) / 2)$. ... | ... - The correct formula for the nth centered hexagonal number is `1 + 3*n*(n-1)`. ... |
| `score` (prob) | $-0.000\,000\,001$ 
 0.499 999 729 ⬤ | $-14.750\,000\,035$ 
 0.000 000 196 ◯ | $-16.750\,000\,053$ 
 0.000 000 027 ◯ | $-14.250\,001\,005$ 
 0.000 000 324 ◯ | $-0.000\,000\,012$ 
 0.499 999 724 ⬤ |
| `solution` | ```def centered_hex... """ ... """ return 3*n*(n-1) + 1``` | ```def centered_hex... """ ... """ if not isinstance(n, int): raise TypeError("Input must be an integer.") if n < 0: raise ValueError("Input must be a non-negative integer.") return 1 + 3 * n * (n + 1)``` | ```def centered_hex... """ ... """ # Calculate the nth centered hexagonal number using the formula return 1 + 6 * (n * (n + 1) // 2)``` | ```def centered_hex... """ ... """ # Calculate the sum of the first n natural numbers sum_of_n = n * (n + 1) // 2  # Calculate the nth centered hexagonal number result = 1 + 6 * sum_of_n  return result``` | ```def centered_hex... return 1 + 3*n*(n-1)``` |
| `score` (prob) | $-0.693\,147\,182$ 
 0.499 999 729 ⬤ | $-15.443\,147\,216$ 
 0.000 000 196 ◯ | $-17.443\,147\,234$ 
 0.000 000 027 ◯ | $-14.943\,148\,186$ 
 0.000 000 324 ◯ | $-0.693\,147\,192$ 
 0.499 999 724 ⬤ |

handle LLM calls. This code has the same structure as the PDL code but requires explicit message building and accumulation (Lines 7-15, and 18-28). It also requires accessing the LLM response data structure (Lines 17 and 30).

Figure 8 presents an implementation of Figure 3 using the IS inference algorithm. The implementation wraps the baseline code in a loop (Line 8) that creates $n$ independent particles. Each particle maintains a log-probability score (Lines 26 and 44) that is used to construct a weighted categorical distribution over solutions (Line 47). This implementation is purely sequential, in particular, each model call is done one after the other.

To implement Figure 3 with SMC, the IS code is restructured as shown in Figure 9. The key difference is that particle execution is split into two phases separated by a resampling step (Line 31): the first loop (Lines 9-29) generates and scores plans, the second loop (Lines 34-54) generates code. This allows SMC to focus computational resources on promising particles after the first scoring point. The `resample` function samples $n$ states from the distribution of particle states.

```
1  def resample(states, scores):
2      n = len(states)
3      d = Categorical(zip(states, scores))
4      return [d.sample() for _ in range(n)]
```

The SMC algorithm can be implemented using prompt programming frameworks like LangChain, as shown in Figure 10. This implementation exploits LangChain's batch API (Lines 21 and 49) to execute particles in parallel, significantly reducing wall-clock time. The batch API sends multiple prompts to the LLM simultaneously.

Compared to these implementations, PPDL provides a high-level abstraction that separates the probabilistic specification from inference implementation details, allowing the runtime to automatically optimize execution (e.g., through parallelism) without programmer intervention.

*Table 5.* Example of execution of the program provided in Figure 3 using SMC. The arrows represents the resampling step. In this execution, all the particles compute a correct solution and they all have the same probability. After the the first `factor`, the resampling step duplicates particles 1 and 2 who have a plan that contains the correct formula.

| Variable | Particle 0 | Particle 1 | Particle 2 | Particle 3 | Particle 4 |
|---|---|---|---|---|---|
| `plan` | ... Simplifying this gives 1 + 3n(n + 1) = 3n^2 + 3n + 1. ... | ... By substituting the sum of the first (n-1) natural numbers into the expression, we get '1 + 6*((n-1)*n/2)', which simplifies to '1 + 3*n*(n-1)'. ... | ... The formula to calculate the nth centered hexagonal number is given by: $1+6+12+...+6(n-1)$, which simplifies to $3n^2-3n+1$. ... | ... Substituting the sum of the first 'n' natural numbers into the formula for the nth centered hexagonal number gives '1 + 6 * (n * (n + 1) / 2)'. ... | ... The formula can be simplified to 1 + 6(n(n+1)/2). ... |
| `score` (prob) | $-10.75$  0.000 010 723 ◯ | $-0.000\,000\,002$  0.499 994 610 ⬤ | $-0.000\,000\,009$  0.499 994 607 ⬤ | $-15.99$  0.000 000 057 ◯ | $-18.75$  0.000 000 004 ◯ |

| Variable | Particle 0 | Particle 1 | Particle 2 | Particle 3 | Particle 4 |
|---|---|---|---|---|---|
| `solution` | ```
def
centered_hex...
"""
...
"""
if not
isinstance(n,
int) or n <= 0:
raise
ValueError("n
must be a
positive
integer")
return 1 +
3*n*(n-1)
``` | ```
def
centered_hex...
return 1 +
3*n*(n-1)
``` | ```
def
centered_hex...
"""
...
"""
# Calculate the
nth centered
hexagonal
number using
the formula:
3n^2 - 3n + 1
return 3*n**2 -
3*n + 1
``` | ```
def
centered_hex...
return 1 +
3*n*(n-1)
``` | ```
def
centered_hex...
"""
...
"""
return 3*n**2 -
3*n + 1
``` |
| `score` (prob) | $-0.69$  0.2 ◯ | $-0.69$  0.2 ◯ | $-0.69$  0.2 ◯ | $-0.69$  0.2 ◯ | $-0.69$  0.2 ◯ |

## C. Semantics

In this section, we present semantics of PPDL.

The syntax of PPDL using the flow-style of YAML is the following:

$$pdl ::= \{\; \texttt{defs:}\{x\texttt{:}pdl, \ldots, x\texttt{:}pdl\},$$
$$body,$$
$$\texttt{parser:}parser,$$
$$\texttt{contribute:}contribute,$$
$$\texttt{def:}x \qquad\qquad\}$$

$$body ::= \texttt{data:}expr$$
$$\mid \texttt{code:}pdl, \texttt{lang:}string$$
$$\mid \texttt{model:}expr, \texttt{input:}pdl$$
$$\mid \texttt{if:}expr, \texttt{then:}pdl, \texttt{else:}pdl$$
$$\mid \texttt{sequence:}[pdl, \ldots, pdl], \texttt{join:}join$$
$$\mid \texttt{while:}expr, \texttt{repeat:}pdl, \texttt{join:}join$$
$$\mid \texttt{function:}types, \texttt{return:}pdl$$
$$\mid \texttt{call:}expr, \texttt{args:}expr$$
$$\mid \texttt{factor:}expr$$

$$expr ::= \texttt{null} \mid bool \mid number \mid string \mid \$\{jinja\_expr\}$$
$$\mid [expr, \ldots, expr] \mid \{x\texttt{:}expr, \ldots, x\texttt{:}expr\}$$

A program is a block comprising a set of variable definitions, the block body (i.e., the instruction to evaluate), a parser to extract the result from the return value of the body, a flag indicating whether the result contributes to the context, and the name of a variable to store this result.

```
1  llm = "watsonx/meta-llama/llama-4-maverick-..."
2  problem_statement = (
3    "Write a function to find nth centered hexagonal number.\n"
4    "assert centered_hexagonal_number(10) == 271\n"
5  )
6
7  solutions_and_scores = []
8  for _ in range(n):
9    score = 0
10   messages = [
11     {
12       "content": (
13         "Generate an English plan for how to generate code "
14         f"for the following problem: { problem_statement }"
15       ),
16       "role": "user",
17     }
18   ]
19   response = completion(model=llm, messages=messages)
20   plan = response.choices[0].message
21   messages = messages + [ plan ]
22   constraint = (
23       "This plan for the following problem is correct.\n"
24       f"{problem_statement}\n"
25   )
26   score = score + utils.llm_judge(llm, plan, constraint)
27   messages = messages + [
28     {
29       "content": (
30         "Generate a complete executable Python function "
31         "definition corresponding to the above plan and "
32         "problem. Generate only a single function definition."
33       ),
34       "role": "user",
35     }
36   ]
37   response = completion(model=llm, messages=messages)
38   solution_str = response.choices[0].message["content"]
39   solution_match = re.fullmatch(
40     r"(.|\n)*```python\n(?P(.|\n)*?)```(.|\n)*",
41     solution_str, flags=re.M
42   )
43   solution = solution_match.group("code")
44   score = score + utils.score_errors_and_warnings(solution)
45   solutions_and_scores.append((solution, score))
46
47 dist = Categorical(solutions_and_scores)
```

*Figure 8.* Implementation in Python using LiteLLM of Figure 3 with IS probabilistic inference.

```
1  llm = "watsonx/meta-llama/llama-4-maverick-..."
2  problem_statement = (
3    "Write a function to find nth centered hexagonal number.\n"
4    "assert centered_hexagonal_number(10) == 271\n"
5  )
6
7  states = []
8  scores = []
9  for _ in range(n):
10   score = 0
11   messages = [
12     {
13       "content": (
14         "Generate an English plan for how to generate code "
15         f"for the following problem: { problem_statement }"
16       ),
17       "role": "user",
18     }
19   ]
20   response = completion(model=llm, messages=messages)
21   plan = response.choices[0].message
22   messages = messages + [ plan ]
23   constraint = (
24       "This plan for the following problem is correct.\n"
25       f"{problem_statement}\n"
26   )
27   score = score + utils.llm_judge(llm, plan, constraint)
28   states.append(messages)
29   scores.append(score)
30
31 states = resample(states, scores)
32
33 solutions_and_scores = []
34 for messages in states:
35   score = 0
36   messages = messages + [
37     {
38       "content": (
39         "Generate a complete executable Python function "
40         "definition corresponding to the above plan and "
41         "problem. Generate only a single function definition."
42       ),
43       "role": "user",
44     }
45   ]
46   response = completion(model=llm, messages=messages)
47   solution_str = response.choices[0].message["content"]
48   solution_match = re.fullmatch(
49     r"(.|\n)*```python\n(?P(.|\n)*?)```(.|\n)*",
50     solution_str, flags=re.M
51   )
52   solution = solution_match.group("code")
53   score = score + utils.score_errors_and_warnings(solution)
54   solutions_and_scores.append((solution, score))
55
56 dist = Categorical(solutions_and_scores)
```

*Figure 9.* Implementation in Python using LiteLLM of Figure 3 with SMC probabilistic inference.

```
1  llm = WatsonxLLM(
2      model_id="meta-llama/llama-4-maverick-...",
3      ...
4  )
5  parser = StrOutputParser()
6
7  problem_statement = (
8   "Write a function to find nth centered hexagonal number.\n"
9   "assert centered_hexagonal_number(10) == 271\n"
10 )
11
12 plan_prompt = HumanMessage(
13   content=(
14     "Generate an English plan for how to generate code "
15     f"for the following problem: {problem_statement}"
16   )
17 )
18
19 plan_inputs = [[plan_prompt]] * n
20 plan_responses = llm.batch(plan_inputs)
21
22 states = []
23 scores = []
24 for response in plan_responses:
25   messages = [plan_prompt, response]
26   plan_text = parser.invoke(response)
27   constraint = (
28     "This plan for the following problem is correct.\n"
29     f"{problem_statement}\n"
30   )
31   score = utils.llm_judge(llm, plan_text, constraint)
32   states.append(messages)
33   scores.append(score)
34
35 states = resample(states, scores)
36
37 code_prompt = HumanMessage(
38  content=(
39    "Generate a complete executable Python function "
40    "definition corresponding to the above plan and "
41    "problem. Generate only a single function definition."
42  )
43 )
44
45 code_inputs = [
46   messages + [code_prompt] for messages in states
47 ]
48 code_responses = llm.batch(code_inputs)
49
50 solutions_and_scores = []
51 for response in code_responses:
52   solution_str = parser.invoke(response)
53   solution_match = re.fullmatch(
54     r"(.|\n)*```python\n(?P(.|\n)*?)```(.|\n)*",
55     solution_str, flags=re.M,
56   )
57   solution = solution_match.group("code")
58   score = utils.score_errors_and_warnings(solution)
59   solutions_and_scores.append((solution, score))
60
61 dist = Categorical(solutions_and_scores)
```

*Figure 10.* Implementation in Python using LangChain of Figure 3 with SMC probabilistic inference.

The body of a block can be an expression (data), some code in Python, Shell, Jinja, or PDL, an LLM call with its input, a conditional, a sequence of blocks with a join operator to gather result values, a loop with a join operator to gather the results of each iteration, a function definition, a function call, or a factor statement to update the score. The *join* operator is a function that takes a list as input and returns a value. In the concrete syntax of PPDL, {join:{as:text}} corresponds to the string concatenation, {join:{as:lastOf}} returns the last element of the list, and {join:{as:array}} is the identity function.

Types (*type*) are a subset of JSON Schema, the parsers (*parser*) are regular expressions, and the contribute flag is defined as follows:

$$contribute ::= [] \mid [\texttt{pdl\_context}]$$

The ideal semantics of a PPDL program $p$ is defined by the infinite number of executions of the weighted sampler associated to the program $p$ and the normalization of the weights to obtain a categorical distribution:

$$\frac{\{p/S \Rightarrow^* v_i/S_i\}_{1 \leq i \leq N} \quad \{w_i = \exp(S_i'[\texttt{pdl\_score}])\}_{1 \leq i \leq N} \quad W = \sum_{1 \leq i \leq N} w_i}{p/S \rightsquigarrow \lambda U. \sum_{1 \leq i \leq N} (w_i/W) \times \delta_{v_i}(U)}$$

The reduction $\Rightarrow^*$ is the transitive closure of the weighted sampler semantics and $N \to \infty$.

The semantics of weighted sampler $p$ is defined Figure 11 by the reduction $p/S \Rightarrow p\_or\_v/S'$ such that, given an environment $S$, the program $p$ evaluates to the updated environment $S'$, and rewrites into a new program or a value.

The semantics of variable definitions is defined Figure 12 by the reduction $defs/S \xrightarrow{defs} defs'/S'$ such that, given an environment $S$, the definitions $defs$ evaluates to the updated environment $S'$, and rewrites into a new set of definitions $defs'$.

The semantics of a block body $b$ is defined Figure 13 by the reduction $b/S \to b\_or\_v/S'$ such that, given an environment $S$, the block body $b$ evaluates to the updated environment $S'$, and rewrites into a new block body or a value.

The semantics of an expression $e$ is defined Figure 14 by the reduction $S \vdash e \Downarrow v$ such that, given an environment $S$, the expression $e$ evaluates a value. The reduction $S \vdash c \Downarrow^l v$ uses the semantics of the language $l$ to evaluate the code $c$ to a value $v$. Evaluation of expressions and external code are not supposed to do side effects.

The semantics of parsing a value $v$ by the parser *parser* to extract a value $v'$ is defined by the semantics of Python regular expressions and is noted $parser, v \xrightarrow{parse} v'$.

The semantics of contribute is defined Figure 15 by the reduction $S, v, contribute \xrightarrow{contrib} S'$ such that, given an environment $S$, the value $v$, and the contribute flag *contribute* produces a new environment $S'$.

## D. Models used in experiments

Table 6 shows the selection of LLMs used in experiments.

*Table 6.* Large language models used in experiments.

| Model | Size | | Dates | |
| --- | --- | --- | --- | --- |
| | total | active | cutoff | release |
| granite4-small | 32b | 9b | 11/2024 | 10/2025 |
| granite4-micro | 3b | 3b | 4/2024 | 10/2025 |
| gpt-oss-120b | 117b | 5b | 6/2024 | 8/2025 |
| gpt-oss-20b | 21b | 4b | 6/2024 | 8/2025 |
| llama4-maverick | 400b | 17b | 8/2024 | 4/2025 |
| llama4-scout | 109b | 17b | 8/2024 | 4/2025 |

## E. Benchmark Code

This section presents the complete PPDL programs used in the evaluation (Section 4). Each program demonstrates how PPDL enables declarative specification of LLM workflows with probabilistic inference through factor statements. The programs use LLM-as-a-judge and rule-based constraints to guide inference toward correct solutions.

### E.1. GSM8k

The GSM8k program (Figure 16) solves grade-school math problems through a simple three-step workflow. First, it prompts the LLM to solve the problem and generate the final answer prefixed by #### (Lines 2-8). Second, it extracts the numerical answer using Python code (Lines 9-16). Third, it applies an LLM-as-a-judge factor (Line 18-28) to score the solution's correctness (Lines 18-28). The fallback attribute (Lines 30-31) assigns a large negative score if an error occurs, effectively filtering out malformed responses during inference.

### E.2. Math500

The Math500 program (Figure 17) follows a similar structure to GSM8k but uses a more sophisticated judging strategy. After generating the solution (Lines 2-5), it uses an LLM-as-a-judge to perform line-by-line reasoning analysis (Lines 6-19), asking the judge to identify any errors in the solution's reasoning. This more detailed validation is appropriate for the harder problems in Math500, which span algebra, geometry, number theory, pre-calculus, and probability. Like GSM8k, it uses a fallback to handle failures.

$$\frac{defs/S \xrightarrow{defs} \{\}/S_1 \quad block\_body/S_1 \rightarrow v/S_2 \quad parser, v \xrightarrow{parse} v' \quad S_2, v', contribute \xrightarrow{contrib} S_3 \quad S' = S_3[x \leftarrow v']}{\{\texttt{defs}:defs, block\_body, \texttt{parser}:parser, \texttt{contribute}:contribute, \texttt{def}:x\}/S \Rightarrow v'/S'}$$

$$\frac{defs/S \xrightarrow{defs} defs'/S' \quad defs' \neq \{\}}{\begin{array}{c}\{\texttt{defs}:defs, block\_body, \texttt{parser}:parser, \texttt{contribute}:contribute, \texttt{def}:x\}/S \\ \Rightarrow \\ \{\texttt{defs}:defs', block\_body, \texttt{parser}:parser, \texttt{contribute}:contribute, \texttt{def}:x\}/S'\end{array}}$$

$$\frac{defs/S \xrightarrow{defs} \{\}/S_1 \quad block\_body/S_1 \rightarrow b/S'}{\begin{array}{c}\{\texttt{defs}:defs, block\_body, \texttt{parser}:parser, \texttt{contribute}:contribute, \texttt{def}:x\}/S \\ \Rightarrow \\ \{\texttt{defs}:\{\}, b, \texttt{parser}:parser, \texttt{contribute}:contribute, \texttt{def}:x\}/S'\end{array}}$$

*Figure 11.* Semantics of a block

$$\frac{}{\{\}/S \xrightarrow{defs} \{\}/S} \qquad \frac{pdl_1/S \Rightarrow pdl_1'/S_1}{\{x_1:pdl_1, x_2:pdl_2, \ldots, x_n:pdl_n\}/S \xrightarrow{defs} \{x_1:pdl_1', x_2:pdl_2, \ldots, x_n:pdl_n\}/S_1}$$

$$\frac{pdl_1/S \Rightarrow v_1/S_1 \quad \{x_2:pdl_2, \ldots, x_n:pdl_n\}/S_1[x_1 \leftarrow v_1] \xrightarrow{defs} defs/S'}{\{x_1:pdl_1, x_2:pdl_2, \ldots, x_n:pdl_n\}/S \xrightarrow{defs} defs/S'}$$

*Figure 12.* Semantics of `defs` field.

### E.3. MBPP

The MBPP program (Figure 18) implements a multi-stage code generation workflow with three scoring points. First, it generates an English plan (Lines 5-12) and scores it using an LLM-as-a-judge (Lines 13-26). Second, it generates Python code based on the plan (Lines 27-49), with regex parsing to extract code from markdown blocks (Lines 41-43). Third, it applies multiple constraints: a rule-based check for function definitions (Line 50), a linter-based score using flake8 warnings (Line 51), and a final LLM-as-a-judge for correctness (Lines 52-66). This demonstrates PPDL's ability to combine different types of constraints—LLM-based, rule-based, and tool-based—in a single program. The program imports utility functions from `utils.pdl` (Lines 1-3).

### E.4. LiveCodeBench

The LiveCodeBench program (Figure 19) adapts the MBPP workflow for more challenging programming problems. It follows the same plan-then-code structure (Lines 5-12 for planning, Lines 27-42 for code generation) but requires the generated function to be named `main` and use `input()` for reading inputs (Lines 30-32), matching the LiveCodeBench format. The program applies three `factor` statements: for the plan (Lines 13-26), for linter warnings (Lines 43-44), and for solution correctness (Lines 45-58).

### E.5. Fever

The Fever fact-checking program demonstrates PPDL's tool-use capabilities through a three-part workflow. Figure 20 defines a Wikipedia search tool using PPDL's `function` declaration (Lines 2-27), which wraps a Python implementation that handles disambiguation and errors. The tool signature is exposed to the LLM (Lines 28-30).

Figure 21 implements the evidence gathering loop. After the LLM generates a response with tool calls (Lines 2-7), the program iterates over each tool call using a `for` loop (Lines 8-58). For each search request, it validates the topic using an LLM-as-a-judge (Lines 28-45) before calling the search tool (Lines 46-48). This prevents wasting API calls on irrelevant searches. The evidence from all searches is joined with newlines (Lines 53-54).

Figure 22 performs the final verification. It first checks whether enough evidence has been gathered using an LLM-as-a-judge (Lines 1-18), then prompts the LLM to determine if the claim is true or false based on the evidence (Lines 19-30). This three-part structure demonstrates how PPDL programs can be modularized across multiple files while maintaining a coherent probabilistic workflow.

$$\frac{S \vdash e \Downarrow v}{\texttt{data:}e/S \to v/S} \qquad \frac{S \vdash p \Downarrow c \quad S \vdash c \Downarrow^l v}{\texttt{code:}p, \texttt{lang:}l/S \to v/S} \qquad \frac{S \vdash model \Downarrow m \quad S \vdash input \Downarrow i \quad sample(m, i) = v}{\texttt{model:}model, \texttt{input:}input/S \to v/S}$$

$$\frac{v = j(vs)}{\texttt{sequence:[]}, \texttt{join:}j^{vs}/S \to v/S} \qquad \frac{p_1/S \Rightarrow v_1/S_1 \quad \texttt{sequence:}[p_2 \ldots, p_n], \texttt{join:}j^{vs+[v_1]}/S_1 \to p'/S'}{\texttt{sequence:}[p_1, p_2 \ldots, p_n], \texttt{join:}j^{vs}/S \to p'/S'}$$

$$\frac{p_1/S \Rightarrow p_1'/S_1}{\texttt{sequence:}[p_1, p_2 \ldots, p_n], \texttt{join:}j^{vs}/S \to \texttt{sequence:}[p_1' \ldots, p_n], \texttt{join:}j^{vs}/S_1}$$

$$\frac{S \vdash c \Downarrow \texttt{true} \quad p_1/S \Rightarrow p_1'/S'}{\texttt{if:}c, \texttt{then:}p_1, \texttt{else:}p_2/S \to p_1'/S'} \qquad \frac{S \vdash c \Downarrow \texttt{false} \quad p_2/S \Rightarrow p_2'/S'}{\texttt{if:}c, \texttt{then:}p_1, \texttt{else:}p_2/S \to p_2'/S'}$$

$$\frac{S \vdash e \Downarrow \texttt{false}}{\texttt{while:}e, \texttt{repeat:}p, \texttt{join:}j^{vs}/S \to j(vs)/S}$$

$$\frac{S \vdash e \Downarrow \texttt{true} \quad p/S \Rightarrow v/S'}{\texttt{while:}e, \texttt{repeat:}p, \texttt{join:}j^{vs}/S \to \texttt{while:}e, \texttt{repeat:}p, \texttt{join:}j^{vs+[v]}/S'}$$

$$\frac{S \vdash e \Downarrow w \quad S' = S[\texttt{pdl\_score} \leftarrow S[\texttt{pdl\_score}] + w]}{\texttt{factor:}e/S \to \texttt{data:}\texttt{""}/S'}$$

$$\frac{}{\texttt{function:}t, \texttt{return:}p/S \to \texttt{function:}t, \texttt{return:}p, \texttt{scope:}S/S}$$

$$\frac{\begin{array}{c} S \vdash f \Downarrow \texttt{function:}t, \texttt{return:}p, \texttt{scope:}S_f \quad S \vdash args \Downarrow args' \quad args' \in t \\ p/S_f[\texttt{pdl\_context} \leftarrow S[\texttt{pdl\_context}], \texttt{pdl\_score} \leftarrow S[\texttt{pdl\_score}]] + args' \Rightarrow v/S_f' \\ S' = S[\texttt{pdl\_context} \leftarrow S_f'[\texttt{pdl\_context}], \texttt{pdl\_score} \leftarrow S_f'[\texttt{pdl\_score}]] \end{array}}{\texttt{call:}f, \texttt{args:}args/S \to v/S'}$$

$$\frac{\begin{array}{c} S \vdash f \Downarrow \texttt{function:}t, \texttt{return:}p, \texttt{scope:}S_f \quad S \vdash args \Downarrow args' \quad args' \in t \\ p/S_f[\texttt{pdl\_context} \leftarrow S[\texttt{pdl\_context}], \texttt{pdl\_score} \leftarrow S[\texttt{pdl\_score}]] + args' \Rightarrow p'/S_f' \\ S' = S[\texttt{pdl\_context} \leftarrow S_f'[\texttt{pdl\_context}], \texttt{pdl\_score} \leftarrow S_f'[\texttt{pdl\_score}]] \end{array}}{\texttt{call:}f, \texttt{args:}args/S \to \texttt{call:function:}t, \texttt{return:}p', \texttt{scope:}S_f', \texttt{args:}args'/S'}$$

*Figure 13.* Semantics of block bodies.

$$\frac{c \in (\texttt{null} \cup bool \cup number\, string)}{S \vdash c \Downarrow c} \qquad \frac{S \vdash e \Downarrow^{jinja} v}{S \vdash \$\{e\} \Downarrow v} \qquad \frac{\{S \vdash e_i \Downarrow v_i\}_{1 \leq i \leq N}}{S \vdash [e_1, \ldots, e_n] \Downarrow [v_1, \ldots, v_n]}$$

$$\frac{\{S \vdash e_i \Downarrow v_i\}_{1 \leq i \leq N}}{S \vdash \{x_1{:}e_1, \ldots, x_n{:}e_n\} \Downarrow \{x_1{:}v_1, \ldots, x_n{:}v_n\}}$$

*Figure 14.* Semantics of expressions.

$$\frac{}{S, v, \texttt{[]} \xrightarrow{contrib} S} \qquad \frac{ctx = S[\texttt{pdl\_context}] + [\{\texttt{content:}v\}] \quad S' = S[\texttt{pdl\_context} \leftarrow ctx]}{S, v, \texttt{[pdl\_context]} \xrightarrow{contrib} S'}$$

*Figure 15.* Semantics of `contribute` field.

```
1  lastOf:
2  - >
3    Question: ${ problem }
4    Reason about this math problem and solve it. Generate
5    the final answer on the last line prefixed by '####'
6  - model: ${ model }
7    def: solution
8    parameters: ${ parameters }
9  - lang: python
10   def: result
11   code: |
12     from pdl.optimize.parse_number import (
13       extract_math_answer
14     )
15     result_line = solution.splitlines()[-1]
16     result = extract_math_answer(result_line)
17
18 - defs:
19     constraint: >
20       Is the solution below to the following math problem
21       correct?
22
23
24       Math Problem: ${ problem }
25
26       Solution: ${ result }
27   factor:
28     ${ stdlib.llm_as_judge(model, constraint, parameters) }
29 - ${ result }
30 fallback:
31  factor: -100
```

*Figure 16.* PPDL program for GSM8k benchmark.

### E.6. MiniF2F

The MiniF2F theorem proving agent demonstrates PPDL's capabilities for complex iterative workflows with external tool verification. This program implements the proof/repair loop described in the case study (Section 4): an LLM generates Rocq/Coq proofs that are verified by the Rocq proof assistant, with errors fed back to guide repair attempts.

Figure 23 shows the initialization phase. The program sets up logging infrastructure (Lines 2-7) and initializes a proof checker that interfaces with the Rocq prover using the rocq-ml-toolbox (https://github.com/LLM4Rocq/rocq-ml-toolbox) (Lines 8-13). It then prompts the LLM with both the formal Rocq statement and an informal description of the theorem to prove (Lines 24-51), providing an example proof format to guide the LLM's output.

Figure 24 implements the core proof/repair loop using a `repeat` block (Lines 1-55). In each iteration, the LLM generates a proof attempt (Lines 8-11), which is parsed to extract the Rocq code (Lines 12-24). The proof is then verified by calling the Rocq checker (Lines 26-28). If verification fails (Lines 30-47), a `factor` of $-1$ is applied (Line 39) to score this attempt, and the error message is fed back to the LLM to guide the next repair attempt (Lines 40-46). This scoring mechanism allows SMC to prioritize particles with fewer errors. The loop continues until either a proof

```
1  lastOf:
2  - "Problem: ${ problem }"
3  - model: ${ model }
4    def: solution
5    parameters: ${ parameters }
6  - defs:
7      constraint: >
8        Consider the solution below to the following math
9        problem. Read it line by line and reason about it
10       to identify any errors in reasoning.
11       If no issues are found, respond True. If any issue
12       is found, respond False.
13
14
15       Math Problem: ${ problem }
16
17       Solution: ${ solution }
18   factor:
19     ${ stdlib.llm_as_judge(model, constraint, parameters) }
20 - ${ solution }
21 fallback:
22  factor: -100
```

*Figure 17.* PPDL program for Math500 benchmark.

is found or the token budget is exhausted (Line 53), with a maximum iteration limit (Line 54).

Figure 25 performs final verification and result collection. After the loop terminates, it double-checks the proof correctness (Lines 7-10) and logs the final status (Lines 11-24). If the proof is complete, it records the proof, the number of steps, and the token usage (Lines 25-36); then raises an exception to signal success and terminate all parallel particles (Line 35). This early termination mechanism is crucial for efficiency: once any particle finds a valid proof, the entire computation stops, avoiding wasted token usage.

This three-part structure demonstrates PPDL's ability to express complex agent workflows with iterative refinement, external tool integration, probabilistic scoring based on intermediate feedback, and coordinated termination across parallel executions.

## F. Case Study: Theorem Proving in Rocq

We presents additional results for the case study.

Figure 26 shows the number of problems solved as a function of proof/repair steps for IS depending on the number of particles. It follows the same trends as SMC (see Figure 6).

Figure 27 shows the number of MiniF2F-Rocq problems solved as a function of the number of tokens. IS and SMC consume $k$ times more tokens to prove the first theorem because they run $k$ particles in parallel, but catch up with the IS@1 strategy which can only explore one trajectory. So the more we increase the token budget, the more the probabilistic approaches are beneficial.

```
1  defs:
2    utils:
3      import: utils.pdl
4  lastOf:
5  - >
6    Generate an English plan for how to generate code for
7    the following problem:
8
9    ${ problem }
10 - model: ${ model }
11   parameters: ${ parameters }
12   def: plan
13 - defs:
14     constraint: >
15       Consider the following plan for generating code for
16       the coding problem below:
17
18       Coding Problem: ${ problem }
19
20
21       Plan: ${ plan }
22
23
24       Is the plan correct?
25   factor:
26     ${ stdlib.llm_as_judge(model, constraint, parameters) }
27 - >
28   Generate a complete executable Python function
29   definition corresponding to the above plan for the prompt.
30   Generate a single function definition only  and nothing
31   else. Do not include docstrings.
32
33
34   ${ problem }
35 - model: ${ model }
36   def: response
37   parameters: ${ parameters }
38 - defs:
39     solution:
40       data: ${ response }
41       parser:
42         regex: (.|\n)*```python\n(?P(.|\n)*?)```(.|\n)*
43         spec: { code: string }
44 - if: ${ solution == None }
45   then:
46     defs:
47       solution:
48         data:
49           code : ${ response }
50 - factor: ${ utils.contains_functions(solution.code) }
51 - factor: ${ utils.eval_number_of_warnings(solution) }
52 - defs:
53     constraint: >
54       Consider the following generated code for the coding
55       problem below:
56
57       Coding Problem: ${ problem }
58
59
60       Code: ${ solution }
61
62
63       Is the solution correct?
64
65   factor:
66     ${ stdlib.llm_as_judge(model, constraint, parameters) }
67 - ${ solution.code | default("")}
```

*Figure 18.* PPDL program for MBPP benchmark.

```
1  defs:
2    utils:
3      import: ../mbpp/utils.pdl
4  lastOf:
5  - >
6    Generate an English plan for how to generate code for
7    the following problem:
8
9    ${ problem }
10 - model: ${ model }
11   parameters: ${ parameters }
12   def: plan
13 - defs:
14     judge_prompt: >
15       Consider the following plan for generating code for
16       the coding problem below:
17
18       Coding Problem: ${ problem }
19
20
21       Plan: ${ plan }
22
23
24       Is the plan correct?
25   factor:
26     ${ stdlib.llm_as_judge(model, judge_prompt, parameters) }
27 - >
28   Generate a complete executable Python function
29   definition corresponding to the above plan for the prompt.
30   Generate a single function definition called `main` only
31   and nothing else. Inputs to the function should be given
32   using `input()`.
33
34   Generate a single function call to the main function.
35
36   ${ problem }
37 - model: ${ model }
38   parameters: ${ parameters }
39   def: solution
40   parser:
41     regex: (.|\n)*```python\n(?P(.|\n)*?)```(.|\n)*
42     spec: { code: string }
43 - factor:
44     ${ utils.eval_number_of_warnings(response=solution) }
45 - defs:
46     judge_prompt: >
47       Consider the following generated code for the
48       coding problem below:
49
50       Coding Problem: ${ problem }
51
52
53       Code: ${ solution }
54
55
56       Is the solution correct?
57   factor:
58     ${ stdlib.llm_as_judge(model, judge_prompt, parameters) }
59 - ${ solution.code | default("")}
60 fallback:
61   factor: -100
```

*Figure 19.* PPDL program for LiveCodeBench benchmark.

```
1  defs:
2    search:
3      description: Wikipedia search
4      function:
5        topic:
6          type: string
7          description: Topic to be searched
8      return:
9        def: result
10       lang: python
11       code: |
12         import warnings, wikipedia
13         warnings.simplefilter("ignore")
14
15         def main(topic: str, *args, **kwargs) -> str:
16           try:
17             return wikipedia.summary(topic)
18           except wikipedia.DisambiguationError as d:
19             return (
20               f"\"{topic}\" may refer to one of {d.args[1]}."
21               "Please retry the search with a more"
22               "specific subject."
23             )
24           except wikipedia.WikipediaException as e:
25             return str(e)
26
27         result = main(topic)
28   tools:
29     data:
30     - ${ search.signature }
```

*Figure 20.* PPDL program for Fever benchmark (Part 1: tool definition).

```
1  lastOf:
2  - ${ problem }
3  - model: ${ model }
4    def: response
5    modelResponse: action
6    parameters: ${ dict(parameters, tools=tools) }
7    parser: json
8  - def: evidence
9    for:
10     tool_call: ${ action.choices[0].message.tool_calls }
11   repeat:
12     text:
13     - if: ${ tool_call.function.name == "search"}
14       then:
15         defs:
16           args:
17             data: ${ tool_call.function.arguments }
18             parser: json
19           topic:
20             text: ${ args["topic"] }
21             fallback:
22               defs:
23                 args:
24                   data: ${ args }
25                   parser: json
26               text: ${ args["topic"] }
27         lastOf:
28         - defs:
29             judge_prompt: >
30               Is the topic below a good topic to search
31               in order to answer the following question:
32
33               Question: ${ problem }
34
35
36               Topic:
37
38               ${ topic }
39
40
41               Respond with only ('true'/'false').
42           factor: >
43             ${ stdlib.llm_as_judge(model,
44                                    judge_prompt,
45                                    parameters) }
46         - call: ${ search }
47           args:
48             topic: ${ topic }
49       else:
50         factor: -100
51       fallback:
52         factor: -100
53     join:
54       with: "\n"
55     fallback:
56       factor: -100
```

*Figure 21.* PPDL program for Fever benchmark (Part 2: evidence gathering).

```
1 - defs:
2     judge_prompt: >
3       Is the evidence below enough to support or refute >
4       the following problem:
5
6       ${ problem }
7
8
9       Evidence:
10
11      ${ response }
12
13      ${ evidence }
14
15
16      Respond with only ('true'/'false').
17    factor:
18      ${ stdlib.llm_as_judge(model, judge_prompt, parameters) }
19 - model: ${ model }
20   parameters: ${ parameters }
21   input: |
22     Consider the evidence:
23     ${ response }
24     ${ evidence }
25
26     Is the following statement true or false?
27     ${ problem }
28
29     Respond with only ('true'/'false').
```

*Figure 22.* PPDL program for Fever benchmark (Part 3: final verification).

```
1  defs:
2    logger:
3      aggregator:
4        file:
5          ${ logdir }/logs_${name}_${pdl_particle_id}.txt
6        flush: true
7        mode: w
8    checker:
9      lang: python
10     code: |
11       from proof_checker import Checker
12       workspace=agent_params.get("workspace", "")
13       result = Checker(workspace=workspace)
14   tokens: 0
15 lastOf:
16 - text: |
17
18     ------------------------------------------------
19     | STARTING PROOF.                              |
20     ------------------------------------------------
21     Goal: ${ formal_statement }
22   contribute: [logger, stdout]
23 - text: |
24     Please provide a proof for the following Rocq/Coq
25         statement:
26
27     ```Rocq
28
29     ${ formal_statement }
30     ```
31     It corresponds to the following informal statement:
32         ```${informal_statement}```
33
34     Please use Coq/Rocq and enclose the proof in triple
35         backticks.
36
37     As an example, if the statement to prove is
38
39     ```Rocq
40     Theorem example (a b : nat) (h : a = b + 1) : a > b.
41     ```
42
43     then your final proof snippet should follow this
44         format:
45
46     ```Rocq
47     From Stdlib Require Import Lia.
48
49     Theorem example (a b : nat) (h : a = b + 1) : a > b.
50     Proof.
51       rewrite h.
52       lia.
53     Qed.
54     ```
55     The last `Rocq` code block will be parsed as your
56         final proof.
57   contribute: [logger, context]
```

*Figure 23.* PPDL program for MiniF2F benchmark (Part 1: initialization and prompt).

```
1  - repeat:
2      text:
3      - text: |
4            ------------------------------------------
5            | [${pdl_particle_id}] ITERATION: ${i}        |
6            ------------------------------------------
7        contribute: [logger, stdout, stderr]
8      - model: ${ model_params["model_id"] }
9        parameters: ${ model_params }
10       def: reasoning
11       contribute: [context, result, logger]
12     - defs:
13         response:
14           data: ${ reasoning }
15           parser:
16             regex:
17               (.|\n)*```(coq|Coq|rocq|Rocq|)\n(?P<rocq
                    >(.|\n)*?)```(.|\n)*
18           spec: { rocq: string }
19       if: ${ response == None }
20       then:
21         defs:
22           response:
23             data:
24               rocq: ${ formal_statement }
25     - defs:
26         result:
27           lang: python
28           code: result = checker.check(response["rocq"])
29         prompt:
30           match: ${ result["status"] }
31           with:
32           - case: Error
33             then:
34               lastOf:
35               - text: |

37                     PROOF ERROR - Attempting to fix ${name}
                          ...
38                 contribute: [logger, stdout]
39               - factor: -1
40               - text: |
41                     Your previous Rocq/Coq proof failed to
                          compile with these verifier
42                     errors:

44                     ${result["error"]}

46                     Please fix your proof.
47                 contribute: [logger, result]
48         tokens:
49           lang: python
50           code: |
51             result = pdl_usage.completion_tokens +
                    pdl_usage.prompt_tokens
52       text: ${ prompt }
53   until: ${ (result["status"] == "Complete") or (tokens >
            agent_params["max_tokens"])}
54   maxIterations: ${ agent_params["max_depth"] }
55   index: i
```

*Figure 24.* PPDL program for MiniF2F benchmark (Part 2: proof/repair loop).

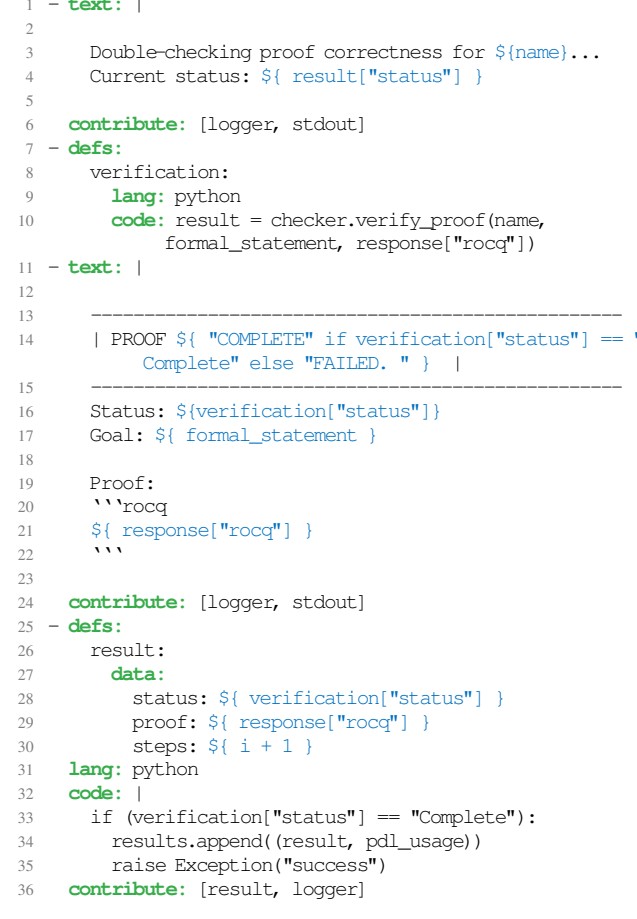

```
1  - text: |
2
3      Double-checking proof correctness for ${name}...
4      Current status: ${ result["status"] }
5
6    contribute: [logger, stdout]
7  - defs:
8      verification:
9        lang: python
10       code: result = checker.verify_proof(name,
                formal_statement, response["rocq"])
11  - text: |
12
13      ------------------------------------------------
14      | PROOF ${ "COMPLETE" if verification["status"] == "
            Complete" else "FAILED. " }  |
15      ------------------------------------------------
16      Status: ${verification["status"]}
17      Goal: ${ formal_statement }
18
19      Proof:
20      ```rocq
21      ${ response["rocq"] }
22      ```
23
24    contribute: [logger, stdout]
25  - defs:
26      result:
27        data:
28          status: ${ verification["status"] }
29          proof: ${ response["rocq"] }
30          steps: ${ i + 1 }
31      lang: python
32      code: |
33        if (verification["status"] == "Complete"):
34          results.append((result, pdl_usage))
35          raise Exception("success")
36    contribute: [result, logger]
```

*Figure 25.* PPDL program for MiniF2F benchmark (Part 3: verification and result collection).

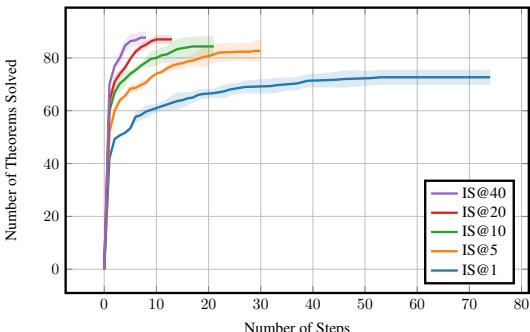

*Figure 26.* Number of MiniF2F-Rocq problems solved as a function of proof/repair steps. Each curve shows the mean over three runs, with shaded regions indicating one standard deviation.

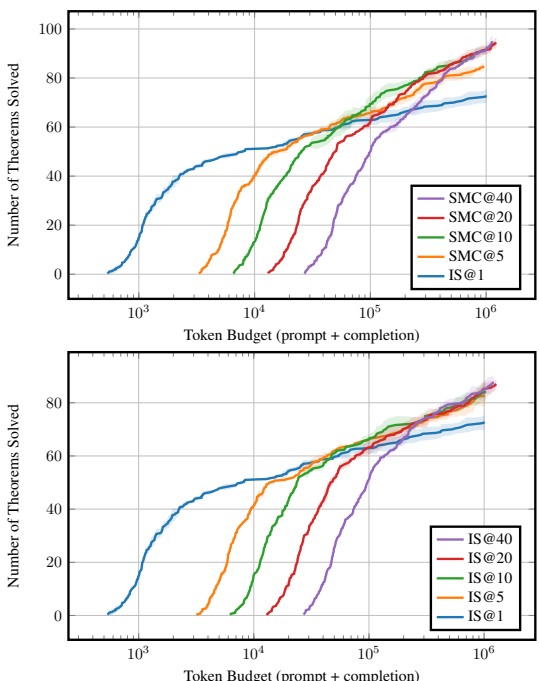

*Figure 27.* Number of MiniF2F-Rocq problems solved as a function of the number of tokens.

