# OpenReview forum: "PPDL: LLM-Based Flows as Probabilistic Programs"
_ICML.cc/2026/Conference — ICML 2026 regular_

### Official Review · Reviewer_KQ4d · 2026-03-04

**Soundness:** 4
**Presentation:** 4
**Significance:** 3
**Originality:** 3
**Overall Recommendation:** 5
**Confidence:** 2

**Summary:**

The authors present PPDL, the first probabilistic programming language for flows of LLM calls and tool calls. The probabilistic inference allows for exploration of the LLM trajectory distributions. The authors evaluate PPDL as inference scaling framework in representative tasks and show that PPDL helps in building inference scaling for downstream tasks.

**Compliance With Llm Reviewing Policy:**

Affirmed.

**Final Justification:**

I keep my current score.

**Key Questions For Authors:**

Since I'm not in the relevant field of this work, I do not have questions to ask for the authors.

**Limitations:**

The authors discuss the limitation of their work.

**Strengths And Weaknesses:**

Thanks for submitting to ICML 2026. I appreciate the authors' contributions to the niche field of applying probabilistic programming with LLMs.

### Soundness

The authors articulate their methodologies clearly with evidence and supported by mathematical formulas.

### Presentation

The text around line 235 - 245 (left) has smaller line spacing.

Section 4.1 also seems like a case study to me, instead of an evaluation of PPDL itself.

### Significance

I anticipate PPDL would help researchers to understand the probabilistic outcome space of programs involving LLM, especially when LLMs being integrated into program's data/control-flow more vastly.

### Originality

The general idea is novel and the usage of PPDL as inference scaling framework is new.

---

> ### Author Rebuttal · Authors · 2026-03-31
>
> We thank the reviewer for the positive assessment and for recognizing the novelty of bringing probabilistic programming to flows of LLM and tool calls, as well as the usefulness of PPDL as an inference-scaling framework. We are especially encouraged that the reviewer sees potential for PPDL to help researchers better understand the probabilistic outcome space of LLM-based programs. PPDL is designed precisely to expose distributions over traces and outputs, rather than only a single execution result, while allowing the same flow to be executed under multiple inference strategies.
>
> Regarding Section 4.1, our intent is to evaluate PPDL as a unified inference-scaling framework, i.e., to show that a single PPDL program can be executed under different inference engines across multiple benchmarks and model families without rewriting the flow logic. The benchmark suite spans reasoning (GSM8K, Math500), code generation (MBPP, LiveCodeBench), and question answering (FEVER), and the results show that the best inference strategy varies across task/model pairs.  We will revise the text to make this framing clearer and to better distinguish Section 4.1 (multi-benchmark evaluation of the framework) from Section 4.2 (the Rocq theorem-proving case study).
>
> Thank you for pointing out the formatting issue around lines 235–245. We will fix the line spacing in the final version.

---

> > ### Author Rebuttal · Reviewer_KQ4d · 2026-04-02
> >
> > Thanks for the reply. I would like to keep the score.

---

### Official Review · Reviewer_DKun · 2026-03-12

**Soundness:** 3
**Presentation:** 3
**Significance:** 3
**Originality:** 4
**Overall Recommendation:** 4
**Confidence:** 3

**Summary:**

The paper introduces PPDL, a probabilistic programming language for LLM-based flows that unifies prompt programming with inference scaling. It extends the prompt language PDL with a single construct, factor, which allows developers to assign log-scores to execution traces and treat multi-step LLM workflows as probabilistic programs. This allows exploration of different inference techinuqes implemented in the language runtime, such as majority voting, importance sampling, and sequential Monte Carlo, without modifying the core program logic. Empirical results  show that inference scaling implemented through PPDL allows to easily  swap inference strategies in agentic workflows and shows comparitive performances, as well as a case study in bulidng a theorem proving agent .

**Compliance With Llm Reviewing Policy:**

Affirmed.

**Final Justification:**

Though concerns around empirical/usability evaluation remain, I keep my positive score 4.

**Key Questions For Authors:**

Please answer any points in the weaknesses section where you can provide more context or information, thanks.

**Limitations:**

yes

**Strengths And Weaknesses:**

Strengths:

- PPDL nicely separates the specification of program logic and workflow structure from the choice of inference strategy (e.g., majority voting, importance sampling, SMC). This separation of concerns provides a nice abstraction layer, potentially allowing developers to scale and experiment with complex LLM or agentic workflows without mixing control logic with sampling  techniques.

- Nice abstraction of "factor" which allows to specify both soft and hard constraints under  a single construct, and can be used to assign higher weight to high-quality execution traces even if they are rare - so weighted inference can prefer a minority but well-scored outcome (according to program-specified preferences) over a simple majority result.

- Evaluation shows that PPDL can serve as a unified runtime for implementing and experimenting with different inference-scaling strategies (majority voting, importance sampling, SMC) across a range of different  tasks, showing that these techniques can be swapped without rewriting the flow and that weighted inference can yield accuracy improvements over single-pass execution.



Weaknesses

- While the paper makes conceptual arguments that PPDL simplifies inference scaling by separating program logic from inference strategy, it does not show empirical evidence that the language improves usability or developer productivity in practice. There is no user study or analysis demonstrating that practitioners can more easily implement or maintain complex LLM flows using PPDL or would prefer it in comparison to their current practice. It is also not clarified who the intended target audience are—e.g. general LLM application developers, ML researchers, or probabilistic programming experts. So it is unclear how accessible and beneficial the language would be for real-world users.

- While PPDL abstracts inference at the level of whole-trace exploration, it is not clear how cases  that require mid-flow aggregation or nested inference would be handled e.g., doing majority voting or Monte Carlo estimation to compute a factor. If this  requires developers to encode those procedures directly within the program logic, then  inference cannot remain truly orthogonal to the model, so it is not clear if the modeling and inference concerns can really be kept separated in all situations in practice.  This raises the question of whether the separation of concerns is as clear-cut as suggested, particularly for more complex agentic workflows.

- The evaluation demonstrates the performance impact of various inference-scaling strategies implemented within PPDL, but the paper does not seem to isolate or quantify the benefits of the language abstraction itself. While efficient and parallelized techniques for implementing inference algorithms in the runtime are presented, the algorithms themselves are standard (IS, SMC), and there is no empirical comparison against existing frameworks or implementations in terms of runtime efficiency, scalability, or cost. So while the integration of these techniques into a unified prompt programming framework is interesting, the evaluation does not clearly establish novel systems-level performance advantages over existing approaches.

---

> ### Author Rebuttal · Authors · 2026-03-31
>
> We thank the reviewer for the constructive feedback. We are encouraged that the reviewer finds the paper technically solid, original, and a meaningful step toward unifying prompt programming and inference scaling. We address below the points raised in the weaknesses section.
>
> 1. Target audience, usability, and developer productivity
>
> The intended target audience of PPDL is developers and researchers who already build multi-step LLM or agentic workflows (e.g., prompt-language users, agent framework developers, and researchers experimenting with inference scaling), rather than probabilistic programming experts. This paper is also targeted to prompt-language and agentic workflow developers who may wish to incorporate probabilistic programming in their tools. PPDL is deliberately designed as a minimal extension of an existing prompt language (PDL), adding a single construct (`factor`) while keeping all inference machinery in the runtime. We will clarify the target audience in the paper.
>
> We agree that the current evaluation does not directly measure developer productivity via a user study. To make the usability more concrete, we will add a side-by-side example in the appendix comparing the programs of Figures 2 and 3 with an implementation in LangChain. In particular, while the Figure 2 flow can be expressed similarly in either framework, the Figure 3 version requires additional cross-cutting logic for particle management, scoring, resampling, distribution construction, and parallelism. This intermixes inference with workflow logic, making the program less declarative and making it harder to swap inference strategies without refactoring. By contrast, in PPDL the flow logic is unchanged and only the runtime configuration selects the probabilistic inference algorithm.
>
> Here is an excerpt of the LangChain code corresponding to Figure 3:
>
> ```
> ...
> plan_prompt = HumanMessage(
>   content=(
>     "Generate an English plan for how to generate code for "
>     f"the following problem: {problem_statement}"
>   )
> )
>
> plan_inputs = [[plan_prompt]] * n
> plan_responses = llm.batch(plan_inputs)
>
> states = []
> scores = []
> for response in plan_responses:
>   messages = [plan_prompt, response]
>   plan_text = parser.invoke(response)
>   constraint = (
>     "This plan for the following problem is correct.\n"
>     f"{problem_statement}\n"
>   )
>   score = utils.llm_judge(llm, plan_text, constraint)
>   states.append(messages)
>   scores.append(score)
>
> states = resample(states, scores)  # resample function from PPDL
> ...
> ```
>
> This code is a short excerpt, and the full version has 70 lines. It is more complicated and error prone than the PPDL implementation, because it has explicit handling of the state and score, calls to `batch`, a `for` loop for each block of code separated by the `factor`s, and `resample` calls.
>
> 2. Mid-flow aggregation and separation of concerns
>
> In the current version of PPDL, mid-flow aggregation is achieved by decomposing the PPDL program into multiple programs that can take the distributions of the previous steps as input. We agree that in this case, the inference does not remain truly orthogonal to the model. Nevertheless, we show that the current approach already improves performance and can be extended with other probabilistic inference algorithms such as MCMC.
>
> 3. Empirical comparison against existing frameworks or implementations in terms of runtime efficiency, scalabilisty, or cost.
>
> We agree that our evaluation does not claim novel inference algorithms or superior runtime performance relative to specialized systems. Our contribution is instead at the language and runtime abstraction level. PPDL allows a developer to write the flow once and execute the same program under different inference strategies without modifying the flow logic itself.
>
> The goal of the empirical evaluation is therefore not to demonstrate efficiency advantages over optimized implementations, but to show that the preferred inference strategy varies across tasks and models, motivating a unified framework in which these strategies are easily interchangeable. We will clarify this framing in the paper so that the evaluation is understood as evidence for the value of the abstraction rather than as a claim of better accuracy or efficiency.
>
> Regarding efficiency, we observed substantial speedups from the parallel implementation. However, measurements exhibited significant variance because experiments were run against shared model-serving infrastructure within a large organization. As a result, we did not include these numbers in the current evaluation. We will clarify this point in the paper and, where appropriate, include additional discussion of parallel speedups in the final version.
>
> More broadly, PPDL is designed to facilitate systematic exploration of the cost–accuracy trade-off by allowing users to switch inference strategies via runtime configuration rather than by rewriting workflow code.

---

> > ### Author Rebuttal · Reviewer_DKun · 2026-04-04
> >
> > Thank you to the authors for the detailed response and further information. I keep my positive score though main concerns around empirical/usability evaluation remain.

---

### Official Review · Reviewer_WmWL · 2026-03-26

**Soundness:** 2
**Presentation:** 1
**Significance:** 1
**Originality:** 3
**Overall Recommendation:** 3
**Confidence:** 5

**Summary:**

The paper introduces PPDL, a probabilistic prompt programming language for LLM-and-tool flows. PPDL extends a declarative prompt language to make LLM flow become a distribution over execution traces instead of a single rollout by leveraging majority voting, importance sampling, and SMC. Case studies on code generation and formal theorem proving shows slight improvement over baseline open-sourced LLMs, yet still fall behind from the state-of-the-art approaches.

**Compliance With Llm Reviewing Policy:**

Affirmed.

**Final Justification:**

I thank the authors for the additional demonstration via the anonymous link, which makes the workflow much clearer and partially addresses some of the concerns. Therefore, I increased my mark to 3.

**Key Questions For Authors:**

N/A. See weakness.

**Limitations:**

Yes, the authors included a limitation paragraph at the end of the paper.

**Strengths And Weaknesses:**

Strengths:

This paper identifies a gap between prompt programming languages and probabilistic programming, and gives a reasonably concrete operational semantics for weighted sampling and approximate inference.

Weakness

1. **Clarity of presenting PPDL:** I found the current Section 2 and 3 extremely hard to read.
In particular, the paper lacks a clear, end-to-end illustrative example that demonstrates how a concrete problem is solved using PPDL. While Figures 2 and 3 provide schematic diagrams of PDL/PPDL programs, they are very dense and difficult to parse on a first reading. Figure 2 and 3 do not show a full execution trace (e.g. intermediate LLM outputs, factor evaluations, and final aggregation), which makes it challenging to understand how the system behaves in practice. Including a step-by-step example  (even a very simple example)  in the main text or appendix would significantly improve the readability.

2. **Related Works:** The related work section does not sufficiently position PPDL with respect to recent prompt programming languages and systems. For example, APPL [1] proposes a programming language for integrating programs and LLM prompts, which appears closely related in spirit. Additionally, widely used systems such as OpenAI Codex CLI and Anthropic Claude Code are also frameworks for multi-step LLM workflows. An explicit comparison will clarify what is fundamentally new in PPDL.

3. **Limitation in Case Study 1.** Part of evaluation relies on benchmarks such as MBPP and LiveCodeBench which are saturated for modern LLMs. As a result, improvements demonstrated on these datasets may not fully reflect performance in more challenging or realistic settings. It would strengthen the empirical section to include more recent and difficult benchmarks (e.g., SWE-Bench[8] or competition-level programming such as IOI problems).

4. **Significant limitation in Case Study 2.** I am mostly concerning the issues of the theorem-proving case study regarding both benchmark selection and interpretation. The paper evaluates on the Coq version of MiniF2F and argues that it is less saturated than the Lean version. However, Lean and Coq differ substantially in terms of libraries and tooling. In particular, Lean 4’s Mathlib provides extensive mathematical formalization that MiniF2F proofs rely on, whereas Coq-based environments are mostly for program verification tasks which lack the comparable support, making direct comparisons meaningless. This discrepancy may partly explain the relatively low absolute performance (66/244 on minif2f with PPDL), and makes it completely non-competitive with SOTA systems such as Hilbert [7] (242/244 on minif2f with Hilbert). Evaluating on alternative benchmarks such as PutnamBench [2], FormalML [3], or LeanGeo [4,5] and comparing against strong baselines (e.g., Hilbert or even weaker self-correction-based methods like Goedel-Prover [6]) would provide a more convincing evaluation, as these datasets are either very hard for LLM (so no contamination concerns) or new.

5.  **Format Issues** in evaluation metrics and results: The definition of Pass@1 in the experimental setup (“repeatedly retry until a proof is found or the token budget is exhausted”) is very different from standard usage in theorem-proving.  Pass@k measures success over at least 1 correct proof over k independent samples, and Pass@1 corresponds to a single attempt. Number of proof correction attempts is orthogonal and different from Pass@k.  Also, Table 2 uses unclear and non-standard abbreviations (e.g., “gr4sm”, “gr4mi”, “maj”), which reduces readability.
Figure 6 is difficult to interpret, with an unusually large number of unnecessary points plotted at similar x-values, making trends unclear.

6. **Code Availability**: The submission does not provide source code in the supplementary material or via an anonymous repository. Given PPDL is partly a new programming system, this will limit the experiment reproducibility and broader use.

Overall, I acknowledge that the authors did discover an interesting research problem and the PPDL has the potential to address it, but the current level of writing and experimental design does not meet the minimum requirement for publication. I hope the authors could carefully address the weaknesses above and make some concrete improvements.

[1] Dong, Honghua, et al. "Appl: A prompt programming language for harmonious integration of programs and large language model prompts." Proceedings of the 63rd Annual Meeting of the Association for Computational Linguistics (Volume 1: Long Papers). 2025.

[2] Tsoukalas, George, et al. "Putnambench: Evaluating neural theorem-provers on the putnam mathematical competition." Advances in Neural Information Processing Systems 37 (2024): 11545-11569.

[3] Yang, Xiao-Wen, et al. "FormalML: A Benchmark for Evaluating Formal Subgoal Completion in Machine Learning Theory." arXiv preprint arXiv:2510.02335 (2025).

[4] Murphy, Logan, et al. "Autoformalizing euclidean geometry." arXiv preprint arXiv:2405.17216 (2024).

[5] Song, Chendong, et al. "LeanGeo: Formalizing Competitional Geometry problems in Lean." arXiv preprint arXiv:2508.14644 (2025).

[6] Lin, Yong, et al. "Goedel-prover-v2: Scaling formal theorem proving with scaffolded data synthesis and self-correction." arXiv preprint arXiv:2508.03613 (2025).

[7] Varambally, Sumanth, et al. "Hilbert: Recursively building formal proofs with informal reasoning." arXiv preprint arXiv:2509.22819 (2025).

[8] Jimenez, Carlos E., et al. "Swe-bench: Can language models resolve real-world github issues?." arXiv preprint arXiv:2310.06770 (2023).

---

> ### Author Rebuttal · Authors · 2026-03-31
>
> We thank the reviewer for their thorough review and thoughtful comments, especially with regards to the case study on the use of AI in Theorem Proving. We would like to note that this paper's main contribution is not about Theorem Proving, rather it introduces a probabilistic programming language for inference scaling that can be applied to any field. To demonstrate its versatility we chose benchmarks to cover a variety of applications including Theorem Proving. In that case study, we show that PPDL can be applied to improve results, illustrating the ease-of-use and flexibility of the programming language.
>
> We appreciate the recognition that the paper addresses an interesting gap between prompt programming and probabilistic programming. We address below each concern in turn.
>
> 1. Clarity of presenting PPDL
>
> We agree that the current presentation of Figures 2 and 3 does not provide an end-to-end execution narrative. To address this, we will add an appendix walkthrough of these examples showing a concrete sampled execution trace, including intermediate LLM outputs, factor values, and the final aggregation into an output distribution.
>
> 2. Related Works
>
> Thanks for pointing out this additional related work. We will cite it. APPL in particular is interesting, since it describes a similar parallelization to what PPDL implements within the execution of a single particle. Note that APPL does not provide probabilistic programming, which is the central contribution of PPDL.
>
> 3. Limitation in Case Study 1
>
> We agree that some benchmarks (especially GSM8K and MBPP) are easier than newer benchmarks. Our goal in Section 4.1, however, is not to claim state-of-the-art task performance, but to evaluate PPDL as a unified inference-scaling framework across heterogeneous tasks and models. To that end, we include reasoning (GSM8K, Math500), code generation (MBPP, LiveCodeBench), and QA (FEVER), and already use LiveCodeBench specifically because it is more recent and contamination-resistant. The main empirical point is that the preferred inference strategy varies across benchmark/model pairs, which motivates a framework where the same flow can be executed under different engines without rewriting the program. We will revise the framing to make this goal more explicit.
>
> 4. Significant limitation in Case Study 2.
>
> We appreciate the reviewer's deep familiarity with formal theorem proving benchmarks and systems and agree that direct numerical comparison with Lean-based systems such as Hilbert is not meaningful due to major differences in libraries and tooling. This case study is here to demonstrate that PPDL can express and execute probabilistic inference strategies in a proof-repair loop with minimal changes to the program. Even if the low baseline performance is coming from Rocq compared to Lean, the point here is to show that the probabilistic approach can improve accuracy. We will clarify absolute performance limitations and remove any language that could be interpreted as competitiveness with specialized theorem-proving systems.
>
> 5. Format Issues
>
> We agree that our use of "Pass@1" is not standard. We will rename this metric as "is@1" which corresponds to execute the PPDL program with importance sampling and one particle. We will also work on the formatting of Table 2 to avoid confusing abbreviations.
>
> 6. Code Availability
>
> PPDL is open-source. The paper currently states that the URL is omitted for double-blind review. We will make the artifact availability statement more prominent and ensure that the code link is provided as well as detailed instructions to reproduce the evaluation.

---

> > ### Author Rebuttal · Reviewer_WmWL · 2026-04-03
> >
> > Thank you for the detailed rebuttal and clarifications.
> >
> > Still, my main concerns remain unresolved. In particular, the theorem-proving case study is still problematic and making nonsense due to the mismatch in benchmarks and lack of meaningful comparisons, which makes the results difficult to interpret and undermines the empirical validity. More broadly, the evaluation design (i.e. benchmark choices and metrics) and presentation issues in Section 4 (particularly the Figure 6 and the format issues) require substantial revision beyond what can be addressed in rebuttal. Promised clarifications like the concrete execution walkthrough were not provided, though the authors are given opportunity to show them in anonymous link according to the instructions in https://icml.cc/Conferences/2026/PeerReviewFAQ.
> >
> > Based on these reasons above, I have to maintain my original rating of 2 and confidence of 5.

---

> > > ### Author Response · Authors · 2026-04-06
> > >
> > > Thank you for pointing out to the rules on figures and table sharing. You can find the following:
> > > - https://anonymous.4open.science/r/figures-D090/figure6.pdf : cleaner version of Figure 6 where we are showing now an average over 3 runs with the standard deviation.
> > > - https://anonymous.4open.science/r/figures-D090/table2.pdf : fix of the formatting in Table 2.
> > > - https://anonymous.4open.science/r/figures-D090/table3.pdf : a new table showing a concrete execution walkthrough.
> > >
> > > We understand the reviewer's concern regarding the choice of Rocq, but we note that even if the ecosystem is not as monolithic as Lean's Mathlib,  there are large-scale math libraries in Rocq (see, e.g., [Mathematical Components](https://github.com/math-comp/math-comp), [MathComp-Analysis](https://github.com/math-comp/analysis), [Coquelicot](https://gitlab.inria.fr/coquelicot/coquelicot), or [GeoCoq](https://github.com/GeoCoq/GeoCoq)) which are the backbone of the Four Color theorem [Gonthier, 2008] and the Feit–Thompson theorem [Gonthier et al., 2013].
> > >
> > > The AI for Math community has mostly focused on Lean and developed specialized models with impressive results. However, recent frontier models coupled with agentic frameworks are now also proficient in Rocq: recent results show that Claude Opus 4.6 was able to formalize [10 out of 12 Putnam 2025 problems in Rocq](https://arxiv.org/abs/2603.20405), and achieves [above 80% on MiniF2F-Rocq](https://rocq-prover.zulipchat.com/#narrow/channel/252087-Machine-learning-and-automation/topic/Opus.204.2E6.20is.20great.20at.20formal.20proofs.20.28Rocq.2FLean4.29/near/575001730) with minimal tooling. This demonstrates that libraries and tooling are not the bottleneck, and that the relatively low performance is due to the limitations of the base model, which is thus a good fit for measuring improvements with inference scaling.
> > >
> > > We emphasize that the goal of this experiment is not to compare our proof agent with state-of-the-art solutions, but to measure how much inference scaling with probabilistic inference can improve the performance of a base model (here gpt-oss-120b) on a challenging task.

---

### Official Review · Reviewer_7L6u · 2026-04-03

**Soundness:** 4
**Presentation:** 3
**Significance:** 4
**Originality:** 3
**Overall Recommendation:** 5
**Confidence:** 2

**Summary:**

The paper introduces a probabilistic programming language for coding LLM flows with tools that allow developers to swap different inference scaling techniques as well as quantify and propagate uncertainty.

**Compliance With Llm Reviewing Policy:**

Affirmed.

**Final Justification:**

I move for acceptance due to solid technical contributions and lean positively regarding soundness, originality, significance, and clarity.

**Key Questions For Authors:**

Overall, I believe this is a strong submission with solid contributions. The following questions are minor clarifications:
1. What is the computational overhead of PDDL relative to writing the inference logic directly?
2. When would importance sampling be expected to work better than majority voting or SMC and vice versa?

**Limitations:**

Yes

**Strengths And Weaknesses:**

Soundness: The paper is technically sound, as it provides formal semantics for weighted sampling and connects it to importance sampling and sequential Monte Carlo. The empirical evidence is sound to my knowledge and spans both benchmarks and a case study.
Presentation: The paper is generally well-written and easy to follow, with clear, motivating examples.
Significance and Originality: The work introduces the first probabilistic programming language for coding LLM flows with tools, which is an important problem, as many LLM-driven applications involve multi-step flows, and tools and uncertainty aren't well captured by conventional approaches.

---

### Decision · Program_Chairs · 2026-04-30

**Decision:**

Accept (regular)

**Comment:**

This paper introduces PPDL, a probabilistic programming language for LLM-based flows that extends a prompt language with a single "factor" construct, enabling developers to treat multi-step workflows as probabilistic programs and swap inference strategies (majority voting, importance sampling, SMC) without modifying flow logic. Evaluations span code generation, reasoning benchmarks, and a theorem-proving case study.

Reviewers recognized the novelty and significance. Reviewer 7L6u stated the paper is "technically sound". Reviewer DKun praised the separation of concerns, noting that "PPDL nicely separates the specification of program logic and workflow structure from the choice of inference strategy."

However, a number of important concerns were raised and not fully addressed in rebuttal. Reviewer WmWL has concerns about the evaluation design (i.e. benchmark choices and metrics) beyond what can be addressed in rebuttal, specifically the lack of meaningful comparisons in the theorem-proving case study. Reviewer DKun had concerns that "no empirical comparison against existing frameworks or implementations in terms of runtime efficiency, scalability, or cost [were presented]".

Although these concerns are not addressing the central claim of the paper, these concerns make it difficult to evaluate the impact and value of the contribution. Therefore I am recommending weak accept.